# Cross-validation of distance measurements in proteins by PELDOR/DEER and single-molecule FRET

Martin F. Peter [1,4], Christian Gebhardt[2,4], Rebecca Mächtel[2], Gabriel G. Moya Muñoz[2], Janin Glaenzer[1], Alessandra Narducci [2], Gavin H. Thomas [3], Thorben Cordes[2✉] & Gregor Hagelueken [1✉]

Pulsed electron-electron double resonance spectroscopy (PELDOR/DEER) and single-molecule Förster resonance energy transfer spectroscopy (smFRET) are frequently used to determine conformational changes, structural heterogeneity, and inter probe distances in biological macromolecules. They provide qualitative information that facilitates mechanistic understanding of biochemical processes and quantitative data for structural modelling. To provide a comprehensive comparison of the accuracy of PELDOR/DEER and smFRET, we use a library of double cysteine variants of four proteins that undergo large-scale conformational changes upon ligand binding. With either method, we use established standard experimental protocols and data analysis routines to determine inter-probe distances in the presence and absence of ligands. The results are compared to distance predictions from structural models. Despite an overall satisfying and similar distance accuracy, some inconsistencies are identified, which we attribute to the use of cryoprotectants for PELDOR/DEER and label-protein interactions for smFRET. This large-scale cross-validation of PELDOR/DEER and smFRET highlights the strengths, weaknesses, and synergies of these two important and complementary tools in integrative structural biology.

[1] Institute of Structural Biology, University of Bonn, Bonn, Germany. [2] Physical and Synthetic Biology, Faculty of Biology, Ludwig-Maximilians-Universität München, Planegg-Martinsried, Germany. [3] Department of Biology (Area 10), University of York, York, UK. [4] These authors contributed equally: Martin F. Peter, Christian Gebhardt. ✉email: cordes@bio.lmu.de; hagelueken@uni-bonn.de

Since the determination of the first macromolecular structures in the 1950s, our knowledge about the structure and function of these molecules has dramatically increased. The protein database (PDB, https://www.rcsb.org) contained more than 190,000 structures at the time of writing this manuscript. In many cases, multiple PDB entries represent the same macromolecule but in different conformations. The latter illustrates the dynamic nature of proteins, i.e., the presence of large- or small-scale structural fluctuations that are often crucial to their biological function[1–4]. Until now, most macromolecular structures were determined by either X-ray crystallography (~90%), nuclear magnetic resonance (NMR, ~10%), or electron microscopy (EM, ~2%) (https://www.rcsb.org). The recent release of the AlphaFold database (https://alphafold.ebi.ac.uk) provided many further, albeit often not yet experimentally verified structures[5]. Undoubtedly, cryo-EM, X-ray crystallography, and AlphaFold predictions can deliver detailed insights into the molecular scaffolds of proteins. Nevertheless, they have the disadvantage that such structures are not determined in liquid solution, but in a crystal lattice, frozen on an EM grid, or even in silico. While the underlying macromolecular dynamics can be inferred by determining multiple structures and combining them into a molecular "movie"[6], this requires additional (biochemical) support to verify the selected order of structural states. Traditionally, the study of such dynamic processes is the strength of NMR. But, NMR is limited to the study of relatively small proteins (typically <70 kDa; larger homo-oligomers are an exception), which renders the analysis of many proteins unfeasible. Due to these limitations, other "integrative" methods have become increasingly popular in the last decade[7–10]. The idea behind the concept of integrative structural biology is to combine models from either of the three classical approaches with data from e.g., hydrogen-exchange mass spectrometry HDX-MS[11], cross-linking mass spectrometry[12], Förster resonance energy transfer (FRET)[13–16], small angle X-ray scattering (SAXS)[17] or electron paramagnetic resonance (EPR) in the form of pulsed electron–electron double resonance spectroscopy (PELDOR, also known as double electron-electron resonance spectroscopy, DEER)[18,19]. These orthogonal techniques allow scientists to study conformational dynamics, to visualize conformational heterogeneity, to derive distance constrains between selected residues, and to determine entire contact interfaces even for heterogenous samples in a near-physiological environment[20]. Such information is often time-consuming or even impossible to obtain with the classical structural biology techniques alone. The hybrid models produced by such integrative approaches can be deposited in the PDB-Dev database[10].

In this study, we tested whether two popular integrative methods, single-molecule FRET (smFRET) and PELDOR/DEER spectroscopy, deliver conclusive and consistent results when they are applied to the same proteins. Both techniques are suitable to determine inter-probe distances at the nanometre scale and can also detect conformational changes of macromolecules in their (frozen) solution state.

The two methods have only rarely been applied to the same macromolecular systems using a large number of identical labelling sites[21–28]. Hence, a systematic comparison of the two methods has been lacking. Here, we provide such a comparative study using the following model proteins: (i) HiSiaP, the periplasmic substrate binding protein (SBP) from the sialic acid TRAP transporter of *Haemophilus influenzae*[29,30], (ii) MalE, also known as MBP (maltose binding protein) from *Escherichia coli*, which plays an important role in the uptake of maltose and maltodextrins by the maltose transporter, MalEFGK$_2$[31,32], (iii) SBD2, the second of two substrate-binding domains that are constituents of the glutamine ABC transporter GlnPQ from *Lactococcus lactis*[33–36] and (iv) YopO from

*Yersinia enterocolitica*, a type-III-secretion system effector protein that is injected into macrophages of the host organism where it becomes activated by forming a tight complex with actin[37,38]. The first three proteins belong to the class of substrate binding proteins (SBPs) from distinct structural SBP categories[39]. It is well established that upon binding of substrate, SBPs undergo a large conformational shift (>10 Å for selected residues) from an open unliganded conformation (apo) to a closed conformation (holo) (Fig. 1a)[39–41]. The virulence factor YopO has a kinase- and a guanine nucleotide

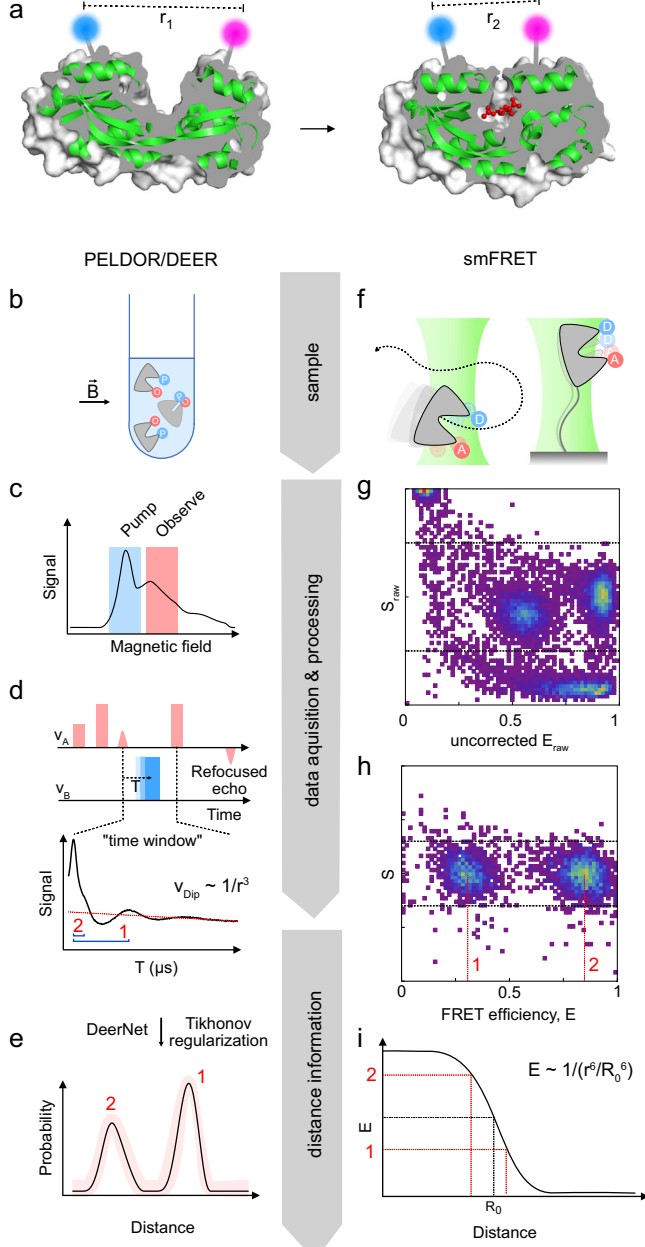

**Fig. 1 Following conformational changes of proteins via PELDOR/DEER or smFRET. a** Two conformations (apo, left and holo, right) of HiSiaP (green cartoon), a substrate binding protein that binds sialic acid (red balls and sticks) (PDB-IDs: 3B50 and 2CEY). A cutaway of the protein surface (grey) is shown to visualize the conformational change of the substrate binding cleft. The position of two labels is indicated by the blue and magenta spheres. **b–e** Workflow of a PELDOR/DEER experiment. **f–i** Workflow of smFRET experiments. The individual steps (**b–i**) are described in detail in the main text.

**Table 1 Common practical questions concerning the applicability of PELDOR/DEER and smFRET to structural biology questions.**

| Structural biology question | PELDOR/DEER | smFRET |
|---|---|---|
| Which range of distances can be measured? | Normally 15–80 Å, but up to 160 Å with fully deuterated proteins | Normally 30–80 Å; 100–150 Å are possible with multiple acceptor dyes |
| How many types of labels are typically needed? | 1 | 2* |
| Can multimeric (homomeric) proteins be studied? | Yes. Multiple distances can be measured in one experiment, using just one type of label and standard equipment. | Yes, but technically demanding. |
| Amount of sample needed? | For Q-band (standard frequency) measurements: 80 µl of ~10–30 µM labelled protein. Measurements with sub-µM spin concentrations have been performed[108]. | 100–400 µl of 15–100 pM labelled protein |
| Physical state of the sample? | Usually frozen solution (50 K). Measurements in aqueous solution are possible but require specific labels (e. g. trityl) and very large or immobilized macromolecules (e.g., ref. [128]). Depending on the label, other pulse sequences than PELDOR/DEER might be required. | Liquid solution at room temperature or cell culture conditions (e.g., 37 °C) |
| In vivo measurements? | Not a standard experiment, especially not under physiological conditions. But, measurements in manually injected frog oocytes or using paramagnetic unnatural amino acids in *E. coli* have been done[59,129]. | Yes |
| Time resolution | Freeze quenched samples can be measured. The time-resolution depends on the freeze quench equipment. Microsecond times have been reached with such equipment[82]. | Down to micro- and nanoseconds |
| Time frame for measurements | Normally several hours per measurement for Q-band | Diffusing molecules: 30–60 min Immobilized molecules: minutes to hours |

*This number does not consider homoFRET approaches where identical fluorophores are used in combination with fluorescence depolarization experiments[130].

dissociation inhibitor (GDI) activity, which both interfere with cytoskeletal dynamics of the host. The crystal structures of the GDI-domain alone and that of the YopO/actin complex have been determined[42,43] and the large conformational changes between apo- and actin bound forms have been studied by PELDOR/DEER in combination with SAXS[44]. The protein is an example of a macro-molecule that can be switched between a presumably dynamic apo state and a rigid ligand-bound state.

All our model proteins are "well-behaved" and have been used previously for either PELDOR/DEER or smFRET experiments. The proteins are therefore well suited for our purpose, i.e., to apply standard procedures for distance determination and then to objectively compare the results. Since PELDOR/DEER and smFRET are often independently used to validate structural models, a cross-validation is important to objectively judge their accuracy, to gauge the severity of their distinct limitations, and to choose the most suitable method for distance measurements for a biological system of interest.

## Results

**A brief comparison of PELDOR/DEER and smFRET.** Although PELDOR/DEER and smFRET are used for similar applications (Fig. 1a), they both have distinct advantages and disadvantages (see Table 1 for a side-by-side comparison). The general workflow of each method is illustrated in Fig. 1 and both methods are briefly described below. We refer the reader to reviews and textbooks (e.g., refs. [45–50]) for a comprehensive description of each methods' theoretical background.

PELDOR/DEER (Fig. 1a–e) is a pulsed EPR experiment that is used to determine the distribution of distances between two or more paramagnetic centres, such as spin labels, which are attached to a macromolecule (Fig. 1b)[45]. At the outset of the experiment, an EPR spectrum of the sample (usually a frozen solution) is recorded to determine suitable microwave frequencies for the pump- and observer pulses (Fig. 1c). By using two different microwave frequencies, it is possible to selectively excite sub-ensembles of the spin centres in the sample, which then serve as either pump spins or observer spins. This explains why only one type of label is needed.

During the actual PELDOR/DEER experiment, the electron spin echo signal ("refocused echo") is produced by the observer pulse sequence (Fig. 1d). If the pump- and observer spins are coupled via a dipole-dipole interaction, the pump pulse sequence (blue square, Fig. 1d) causes characteristic oscillations of the refocused echo. A plot of its intensity for different times T is called PELDOR/DEER "time trace" (Fig. 1d, bottom). The oscillation frequencies of the time trace encode the magnitudes of the dipolar coupling constant that is inversely proportional to the third power of the inter-spin distance (blue brackets in Fig. 1d, bottom). Once recorded, the time-traces can be converted into distance distributions (Fig. 1e), either by applying Tikhonov regularization[51,52] or by using artificial neural networks such as DeerNet that were trained on large sets of simulated data[53]. While PELDOR/DEER is by far the most used experiment for pulsed EPR distance measurements, other pulse sequences have been developed (DQC, SIFTER, RIDME[54–56]). Their applicability to a particular system of interest depends on the spin centres that are used in the experiment (see below).

Distance determination via smFRET (Fig. 1a) is performed in liquid solution on single diffusing or immobilized molecules (Fig. 1f) and relies on the dipole-dipole coupling between two spectrally distinct fluorophores to determine the efficiency of non-radiative energy transfer from the electronically excited donor to the acceptor (Fig. 1g). The energy transfer efficiency depends on the presence of isoenergetic transitions in both molecules (donor emission and acceptor absorption), their relative orientation and the distance between the fluorophores and must be corrected for setup-dependent parameters when a ratiometric, i.e., intensity-based approach is used (Fig. 1h). Accurate FRET efficiencies E can then be converted into distances r using the Förster equation (Fig. 1i), which requires knowledge of the Förster radius $R_0$ (inter-probe distance with $E = 0.5$). An overview of correction parameters for conversion of setup-dependent to accurate FRET efficiencies and distances is provided in Supplementary Table 3.

Because most proteins are diamagnetic and devoid of any suitable fluorophores, both PELDOR/DEER and smFRET experiments usually require the attachment of spin- or fluorescence labels

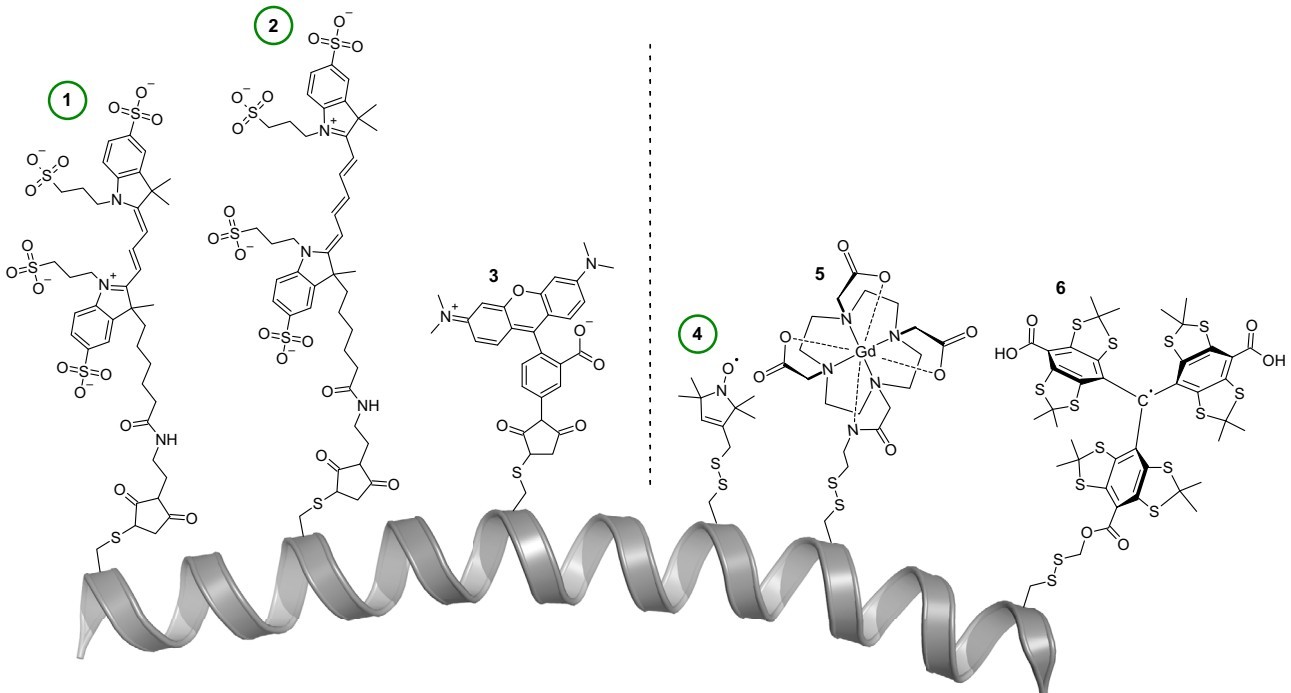

**Fig. 2 Chemical structures of labels commonly used for smFRET (left) and PELDOR/DEER (right).** Green circles identify the commonly used probes for the two methods. **(1)** Maleimide-thiol adducts of Alexa Fluor 555[72] **(2)** Alexa Fluor 647[72] were used in the present study. **(3)** TMR (tetramethylrhodamin-5-maleimide)[131] and **(4)** The MTSSL-[70], **(5)** DOTA-Gd-[132], and **(6)** trityl- spin labels[106,107] attached to a cysteine residue via a disulfide bridge. The polypeptide chain is represented as a grey cartoon. Note that many other labels with differing coupling chemistries ranging from click-chemistry to unnatural amino acids have been developed, as well as labels that are specific for nucleic acids.

(Fig. 1). This can be accomplished by the site-specific introduction of cysteines. The sulfhydrylgroups of cysteines can be reacted with functionalized labels containing maleimides or thiosulfate esters (see Fig. 2 for some examples). If the introduction of cysteines is not possible, alternative labelling approaches such as labelled nanobodies[57] or unnatural amino acids can be used[58–61]. The latter can either be fluorescent or paramagnetic themselves or bear functional groups that can be labelled, for instance by click-chemistry[58–61]. Although the types of labels used for PELDOR/DEER and FRET are quite different, the requirements for suitable labelling positions in proteins are essentially the same: the residue should be solvent-accessible and its labelling should not impair folding or functional properties. For PELDOR/DEER spectroscopy the distance between the labels should be in the range of 1.5–8.0 nm (longer distances of up to 16 nm are however accessible with fully deuterated samples)[45,62,63]. The ideal distance for FRET experiments is around the Förster radius of the selected donor-acceptor pair. This provides a typical dynamic range between 3 and 8 nm (Fig. 1), but in principle, also longer distances up to 10–15 nm are accessible[64]. Usually, labelling positions are chosen such that the distance change between conformations is as large as possible. In practice, the pool of suitable labelling sites is often surprisingly small. Fortunately, software programmes exist to assist in the identification of optimal labelling positions in the case of an available structure or model of the target protein[65–69].

Our aim for the comparison was to choose commonly used labels and well-established experimental conditions for either method. The PELDOR/DEER experiments were thus performed at 50 K with cryo-protected samples using a commercial pulsed Q-band spectrometer. The samples were labelled with MTSSL (S-(1-oxyl-2,2,5,5-tetramethyl-2,5-dihydro-1H-pyrrol-3-yl)methyl methanesulfonothioate)[70]. The distance distributions were determined using Tikhonov regularization[52] and DeerNet[53] and the distance predictions were calculated with mtsslWizard[71]. The

smFRET experiments of diffusing protein molecules were conducted in buffer at room temperature using standard procedures suitable for microsecond alternating laser excitation (ALEX) as described before[72]. The experiments were performed with Alexa Fluor 555 (donor) and Alexa Fluor 647 (acceptor) using a homebuilt confocal microscopy setup with 2-colour excitation/detection[72].

**Comparison 1: Sialic acid binding protein HiSiaP.** Figure 3a shows a difference distance map of HiSiaP, based on the open- and closed crystal structures. The map represents all distance changes between the Cβ atoms of the two states. We picked pairs of sites with pronounced distance changes (dark areas) of up to 1.8 nm for labelling. Figure 3b shows the open (PDB-ID: 2CEY) and closed structures (PDB-ID: 3B50) of the protein with the predicted accessible volumes of the spin- (magenta) and FRET-(blue) labels at the selected labelling sites (residues 55, 58, 134, 175, and 228). For PELDOR/DEER, all double variants (58/134, 55/175, 175/228, and 112/175) were labelled with MTSSL (Supplementary Fig. 1). In each case, two PELDOR/DEER measurements were performed, one in the presence and one in the absence of sialic acid (Neu5Ac). Similar to the previously published PELDOR/DEER data for the *Vibrio cholerae* homolog VcSiaP[73], which shares 49% amino acid sequence identity and 69% sequence similarity with HiSiaP, the PELDOR/DEER-time traces obtained for HiSiaP were of excellent quality with clearly visible oscillations and high signal to noise ratios (SNR, Supplementary Fig. 2). The distance distributions had a single, well-defined peak (Fig. 3c, black curves) with very small uncertainties (red shades) and a clear shift towards shorter distances in the presence of substrate. The corresponding in silico predictions, based on the crystal structures (Fig. 3c, grey areas), were in good agreement with the experimental PELDOR/DEER results, considering the known error of ± 2–4 Å for such predictions. This error margin is mainly due to difficulties in correctly predicting

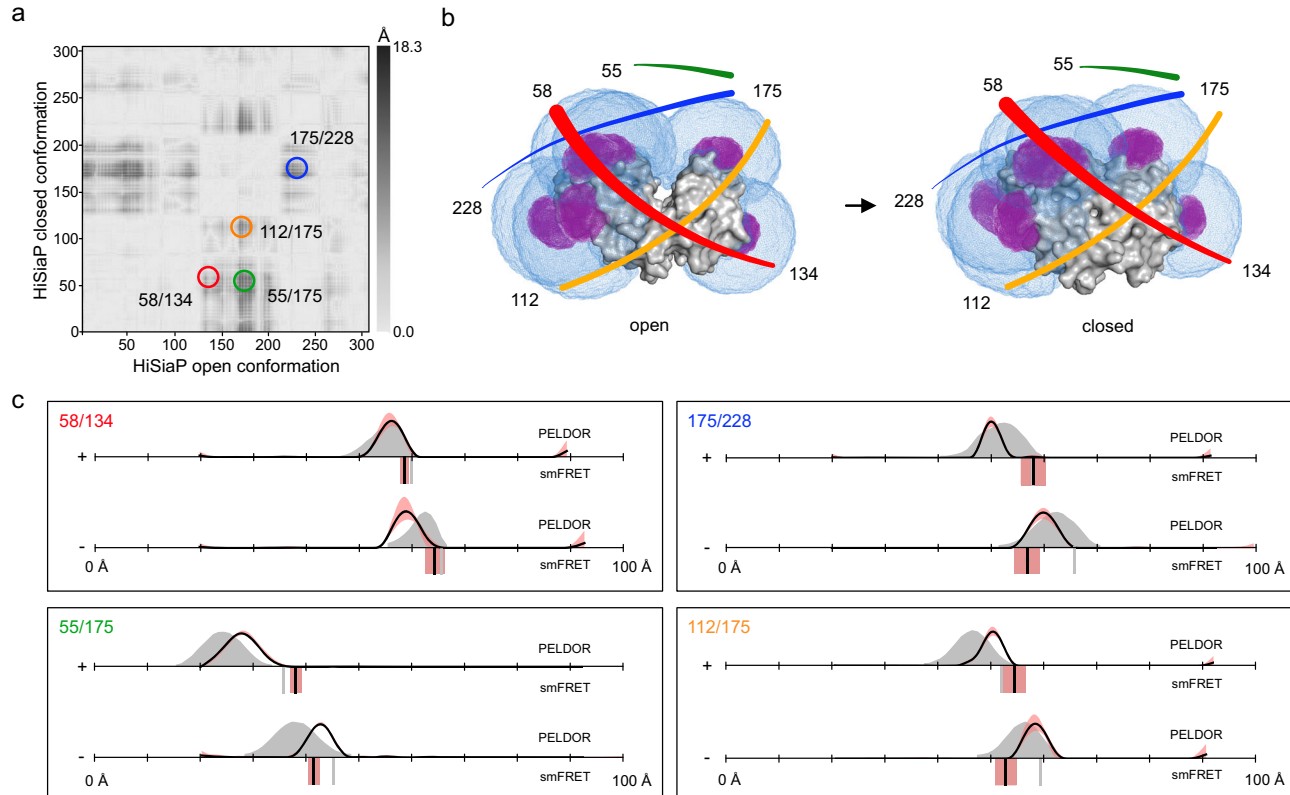

**Fig. 3 Distance measurements on HiSiaP. a** Difference distance map of HiSiaP in the open (PDB-ID: 2CEY[29]) and closed (PDB-ID: 3B50[133]) conformation[125]. The dark spots mark protein regions that undergo large conformational changes relative to each other. The double variants for distance measurements are highlighted with circles. **b** Surface presentation of HiSiaP (grey) in the open (left) and closed (right) conformation. The accessible volumes of the spin label at six different labelling positions were calculated with mtsslWizard and are represented by magenta meshes. Accessible volumes of FRET label maleimide-Alexa Fluor 647 were calculated with FPS[67], and are shown as blue meshes. The double variants that were used for experiments are identified with coloured lines, corresponding to the circles in (**a**). **c** Distance measurements with four different double variants of HiSiaP without (−) and with (+) substrate. The PELDOR/DEER results are shown above (grey curves for simulation, black curves for experiment) and the FRET distances below the x-axis (grey bars for simulation, black bars are the mean of n = 3 independent experiments). Raw data for all experiments are provided in the Supplementary Information. The red shade around the PELDOR/DEER data is the error margin calculated using the validation tool of DeerAnalysis[52]. The underlying principle of the validation tool is explained in the discussion section below. The red shades around the experimental smFRET distances are error bars based on the standard deviation of n = 3 independent experiments. Source data are provided as a Source Data file.

the conformation of the spin label, as discussed below and in[71]. In summary, for each double variant, distance changes were measured that were similar in magnitude to those calculated from the crystallographic models, and in agreement with those from the previously published VcSiaP data[73].

We next assessed substrate-induced conformational changes in the HiSiaP double variants by smFRET spectroscopy using Alexa Fluor 555 and Alexa Fluor 647 as donor and acceptor dye, respectively (Fig. 2). This popular FRET pair was chosen for its high photostability, signal intensity and proven compatibility with various protein samples[33,74,75]. Labelling quality and sample purity were assessed by size exclusion chromatography (SEC) (Supplementary Fig. 3). All samples showed high labelling efficiencies (>90%) and donor-to-acceptor labelling ratios up to ~1:1. The experiments were conducted with freely diffusing molecules at ~50 pM concentration to derive mean FRET-efficiency values for the apo- and holo states of HiSiaP. All smFRET measurements gave high quality ES-histograms with clearly defined populations (Supplementary Fig. 4). However, some of the donor-acceptor populations appeared to be broader than expected, indicating that they were either composed of molecules with conformational flexibility, or there were unwanted photophysical effects arising from the choice of fluorophores and labelling positions (see "Discussion" below).

Figure 3c summarizes the FRET distance measurements in direct comparison with the PELDOR/DEER distance distributions. For smFRET (black bars) only variants 58/134 and 55/175 gave the expected trend for shorter inter-probe distances in the presence of ligand. The two other variants (175/228, 112/175) failed to reproduce the trends from PELDOR/DEER and structural predictions (grey bars). Instead, smFRET data suggested that the apo protein adopted a conformation that was "more closed" than the substrate-bound conformation. Since variants 58/134 and 55/175 agreed with both the PELDOR/DEER results and the models based on the crystal structures, we thought it unlikely that a completely unexpected structural feature of the protein was responsible for the observed discrepancies. Considering the known ± 5 Å experimental accuracy of FRET[76], one might argue that the two states were simply not discernable for the "offending" double variants, (contradicting the simulation results in Fig. 3c). To examine whether HiSiaP was undergoing FRET dynamics in the apo state, we carried out a burst-variance analysis (BVA)[77] of three HiSiaP variants (Supplementary Fig. 5). In BVA the shot-noise-limited standard deviation for a given mean FRET efficiency (STD of FRET, Supplementary Fig. 5) is compared against the experimental standard deviation within the burst related to statistical noise (Supplementary Fig. 5, black solid half circle). The results showed that the protein exists in stable FRET states and does not switch

rapidly between distinct states (on the millisecond timescale). As an example, a DNA hairpin structure undergoing such changes is shown in Supplementary Fig. 5. These results however do not rule out transitions on a timescale below ~500 μs and the characterization of such dynamics would require pulsed interleaved excitation (PIE) or multiparameter fluorescence detection (MFD) analysis[78]. A possible explanation for the discrepancy between crystal structures and smFRET distances was that the fluorescence labels were partly immobilized by an interaction with a surface feature of the protein. For instance, the sulfonic acid groups of the fluorophores could interact with positively charged patches on the protein surface. Because these effects are highly location dependent, the stochastic labelling combination of donor-acceptor pair and acceptor-donor pair might result in a heterogenous mix of two different types of labelled proteins, where the fluorophores "feel" a different environment depending on the environment of either cysteine. Interestingly, we could observe such a broadened population[79] caused by these two labelling combinations very clearly for a distinct dye-combination (Alexa Fluor 546 – Star 635P) for variant 112/175 (Supplementary Fig. 6). Notably, we found this behaviour of broadened apo populations for many HiSiaP variants suggesting fluorophore interactions with the protein surface (Supplementary Fig. 4 and Supplementary Table 1).

To explore the possibility of such unwanted dye-protein interactions, we investigated fluorescence anisotropy and lifetime decays of labelled HiSiaP for two different amino acid positions with a variety of fluorophores. FRET remains a reliable distance ruler for the scenario that at least one of the two fluorophores undergoes free rotation, which is characterized by fast decay of initial anisotropy values and low residual anisotropies. Because the smFRET results for the variant 58/134 were in good agreement with the simulations, we selected the single variant at position 58 as a positive control, where we expected low dye-protein interactions. And because position 175 was involved in both double variants that showed unexpected mean FRET values and broad FRET distributions, particularly in the apo state, the single variant at this position was chosen as a likely negative example.

The anisotropy decays observed for Alexa Fluor 555 revealed interactions of the dye with the protein at both positions (175 and 58) in the apo/holo states with residual anisotropies at long delay times just below 0.3 (Fig. 4a). Yet, Alexa Fluor 555 was fully immobile in the apo conformation for residue 58 (Fig. 4a, apo). Here, we could not identify a short decay component of the time-resolved anisotropy signal on the timescale of fluorophore rotation <1 ns. For Alexa Fluor 647, both positions showed smaller residual anisotropies (Fig. 4b), but also revealed a distinction between apo- and holo state for position 175. In addition, slower anisotropy decays were accompanied by an increase in the fluorescence lifetime of the fluorophores, a finding that was more pronounced for Alexa Fluor 555 than for Alexa Fluor 647 (compare apo/holo decays at position 175, Fig. 4).

The described photophysical effects with fluorophores at position 175 can explain two observations in our measurements: Firstly, the larger lifetime increases of Alexa Fluor 555 compared to Alexa Fluor 647 would lead to a broadening of the FRET distribution, because having either the donor or the acceptor at position 175 would result in two distinct FRET states. Secondly, the changes in lifetime (quantum yield), orientation, and fluorophore disposition experimentally change the Förster radius and thus impact the proper conversion of FRET-efficiency to distance. To avoid these problems, we altered the FRET fluorophore pair to TMR and Cy5 and repeated the full set of experiments. In contrast to Alexa Fluor 555 and Alexa Fluor 647, these fluorophores are not negatively charged and the linker of TMR is significantly shorter as compared to Alexa Fluor 555 (Fig. 2). The TMR/Cy5 label pair was not our first choice, because

it is inferior to Alexa Fluor 555/647 in terms of signal intensity and photostability. Also, for many proteins, charged labels are known to be less prone to stick to the protein surface than hydrophobic labels. In anisotropy and lifetime measurements on the 175 variant in apo and holo state (Fig. 4c), however, TMR showed almost ideal behaviour with high rotational freedom and a fluorescence lifetime that was unaffected by the conformational state of HiSiaP (Fig. 4c). Cy5 revealed a slightly longer fluorescence lifetime in the apo state. In smFRET experiments using variant 175/228 with TMR/Cy5, this reduced fluorophore-protein interaction resulted in qualitative consistency between FRET-derived and simulated distances (Fig. 4d, e).

Taken together, this meant that with the TMR/Cy5 fluorophore pair, the expected substrate-induced domain closure was observed (Fig. 4d, e), and the absolute distance measurements were in much better agreement with the crystal structures. Thus, it appears that the unexpected FRET results with 175/228 and 112/175 were indeed due to interactions of both donor and acceptor fluorophores with the protein surface at this position. The fact that the 55/175 variant behaved as expected shows that protein label interactions do not necessarily lead to wrong results, because the relative orientation of the two labels is also important. It should be noted that also for spin labels, intricate protein/label interactions are known to occur and have been shown to explain initially puzzling results[80].

**Comparison 2: Maltose binding protein MalE.** MalE has previously been studied by smFRET using Alexa Fluor 555 and Alexa Fluor 647 to elucidate the transport mechanism of maltose ABC importer MalFGK$_2$-E (Supplementary Figs. 7 and 8)[75]. In the present work, we used four MalE cysteine double variants (87/127, 36/352, 29/352, and 134/186) with distinct ligand induced distance changes; see difference distance matrix in Fig. 5a. Variants 36/352, 29/352 were designed to show a decrease of distance upon maltose addition. For the 87/127 variant, the labels were located on the opposite surface of the protein to the substrate binding site near the hinge region, and therefore the expectation from the crystal structures was that the distance would increase for the holo state compared to the apo state. We also included a negative control (134/186), in which the two labels were in the same domain of the protein (Fig. 5c) with the expectation that no substrate-induced distance change occurs (Fig. 5c). All variants showed very good agreement between experimental and simulated values in smFRET experiments. Ligand binding was confirmed for all variants using microscale thermophoresis[72].

PELDOR/DEER distance measurements were performed on the same set of variants using MTSSL (Fig. 5 and Supplementary Fig. 1). The phase memory times of the MalE samples were significantly shorter than for the HiSiaP samples (Supplementary Fig. 17). Nevertheless, it was possible to measure time traces with sufficient length to resolve the expected distances, albeit at a lower SNR compared to the HiSiaP samples (see Discussion and Supplementary Fig. 9). Three of the four MalE variants yielded good-quality PELDOR time traces. The 87/127 variant had a relatively low modulation depth and SNR but still provided data of sufficient quality. For all variants, except 29/352, the measured distances closely matched the predictions obtained for the crystal structure of the apo form of the protein (Fig. 5c). Notably, this variant also had the worst match between the simulation and experiment for the smFRET experiments. The addition of 1 mM maltose to all four variants had different effects on the corresponding PELDOR distance distributions. As expected, the ligand did not significantly change the position of the distance peak for 134/186 variant (our negative control). The distance distribution obtained for the variant 87/127 suggested that this variant

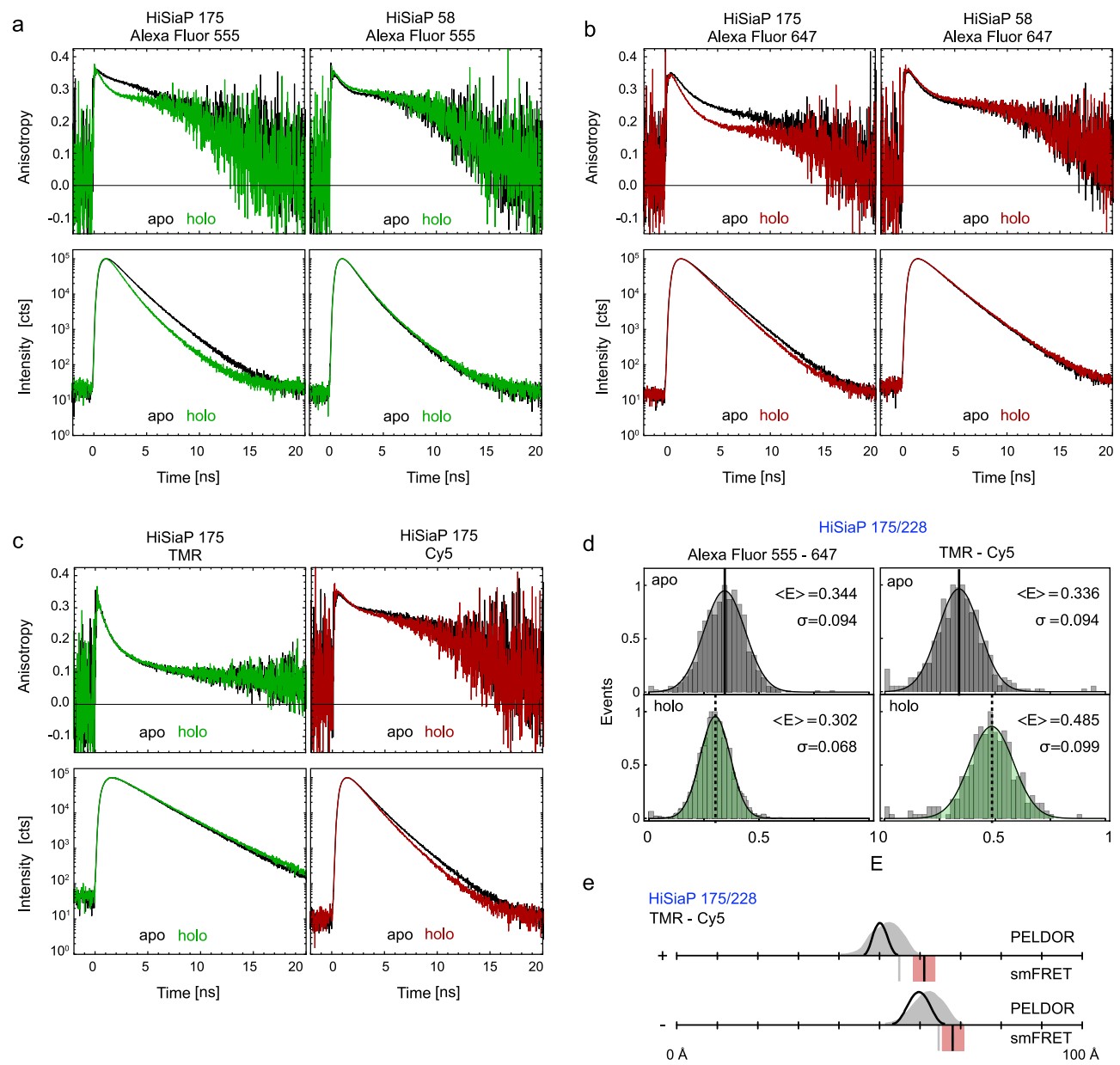

**Fig. 4 Time resolved fluorescence anisotropy and lifetime measurements on HiSiaP. a** Anisotropy decay curves of Alexa Fluor 555 (top row) at residue 175 (left) and 58 (right) and lifetime decay curves (bottom row) under magic angle conditions for apo (black) and holo state (green). **b** Same measurements as in (**a**) with Alexa Fluor 647 in apo (black) and holo state (red). **c** Anisotropy decay curves of TMR (top left) and Cy5 (top right) at residue 175 and lifetime decay curves of TMR (bottom left) and Cy5 (bottom right) under magic angle conditions for apo (black) and holo state (coloured). **d** FRET efficiency distributions (centre & bottom) of HiSiaP variant 175/228 for Alexa Fluor 555 – Alexa Fluor 647 (left) and TMR – Cy5 (right) in apo (grey) and holo state (green). **e** Converted distances from the mean FRET efficiencies are shown as black bars (mean of $n = 3$ independent experiments) in comparison to simulation (grey bar) and PELDOR/DEER results from Fig. 3. The red shade around the experimental smFRET distance are error bars based on the standard deviation of $n = 3$ independent experiments. Source data are provided as a Source Data File.

adopts a closed state. However, the distance distributions obtained for the variant 36/352 appeared as a mixture of the holo- and apo states. A similar result was obtained for variant 29/352 with the only difference that the distance assigned to the apo state was ~5 Å longer than the one obtained from the experiment without maltose. Since the SNR of the PELDOR/DEER time trace for the 87/127 variant was low and the distance change between open and closed conformation was relatively small, we cannot exclude the possibility that this variant also existed in an open-closed mixture after substrate addition (Supplementary Fig. 9).

Because binding constants are temperature dependent, and the PELDOR/DEER samples were measured in the frozen state, we

checked, whether complete closure of MalE was achievable at a higher substrate concentration of 10 mM maltose. Within error, these experiments yielded the same mixtures of the holo- and apo states as seen with 1 mM maltose (Supplementary Fig. 9). Since the lack of complete closure did not result from a sub-saturating maltose concentration in the frozen samples, and it was not observed in the smFRET data, we reasoned that perhaps the cryoprotectant used for PELDOR/DEER experiments might influence the results. Figure 5c shows the PELDOR/DEER results for variants 36/352 and 29/352 in the presence of 1 mM maltose, and in the presence (black lines) or absence (magenta lines) of 50% ethylene-glycol cryoprotectant. Further measurements with 25%

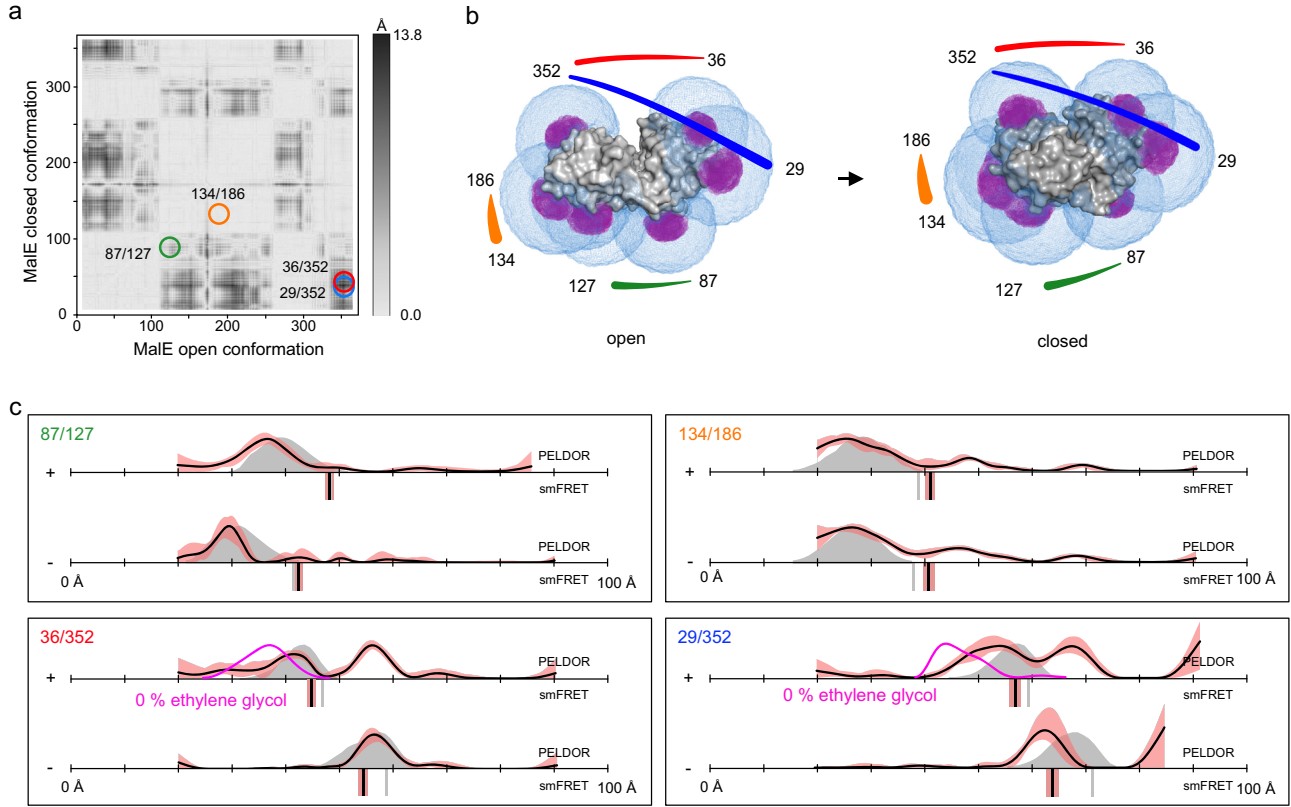

**Fig. 5 Distance measurements on MalE. a** Difference distance map of MalE in the open (PDB-ID: 1OMP[134]) and closed (PDB-ID: 1ANF[135]) conformation[125]. Protein regions with high conformational changes are indicated as dark spots. The double variants for distance measurements are marked with circles. **b** Surface presentation of MalE (grey) in the open (left) and closed (right) conformation. The accessible volume of the spin label on seven different labelling positions, calculated with mtsslWizard, is represented by magenta meshes. The accessible volume of FRET label maleimide-Alexa Fluor 647, calculated with FPS[67] is shown as blue meshes. The double variants that were used for experiments are identified with coloured lines, corresponding to the circles in (**a**). **c** Distance measurements with four different double variants of MalE without (−) and with (+) substrate. The PELDOR/DEER results are shown above (grey curves for simulation, black curves for experiment) and the FRET distance below the axis (grey bars for simulation, black bars are the mean of $n = 3$ independent experiments). PELDOR/DEER results without cryoprotectant are shown as magenta curves. The red shade around the PELDOR/DEER data is the error margin calculated using the validation tool of DeerAnalysis[52]. The underlying principle of the validation tool is explained in the discussion section below. The red shades around the experimental smFRET distances are error bars based on the standard deviation of $n = 3$ independent experiments. Source data are provided as a Source Data file.

ethylene-glycol or glycerol are given in Supplementary Fig. 10. Interestingly, with 25% glycerol, the assumed closed state of MalE became more dominant and in the absence of cryoprotectant, it was the only state that was detected. Note that for the measurements without cryoprotectant, the length of the PELDOR/DEER time traces had to be shortened to achieve a sufficient SNR. Inevitably, this made the corresponding distance distributions less reliable for longer distances (see "Discussion"). However, Supplementary Fig. 10 shows that also the oscillation periods of the time traces themselves were different, clearly revealing the influence of the cryoprotectant on our measurements.

In summary, for MalE, both methods were able to detect the substrate-induced closure of the protein. A reasonable consistency between the two methods and also in relation to structure-based predictions was found. The cryoprotectant used in PELDOR/DEER measurements was identified as the source of initial discrepancies.

**Comparison 3: Glutamate/Glutamine binding protein SBD2.** Previously to this work, we studied the SBD2 domain of the GlnPQ amino acid transporter by smFRET spectroscopy to elucidate its binding mechanism and its involvement in amino-acid transport[33]. Here, we conducted smFRET experiments to

determine accurate FRET efficiencies for variants 319/392 and 369/451 using Alexa Fluor 555 and Alexa Fluor 647 (Supplementary Figs. 11 and 12). The experimental FRET distances were in good agreement with the in silico predictions (Fig. 6c), where both apo and holo proteins showed a single population. In contrast, in the PELDOR/DEER experiments, the apo form of both SBD2 variants displayed at least two prominent distance peaks (Fig. 6c and Supplementary Fig. 13).

The in silico predictions (grey curves) indicated that these distance peaks can be assigned to the open- and closed conformations. After addition of the substrate glutamine to the protein sample, the relative ratio between the two PELDOR/DEER distances shifted towards shorter distances. Thus, the closed conformation became dominant, but at least ~10% of the protein remained in the open conformation (Fig. 6c). In this case and within error, the removal of the cryoprotectant had no significant effect on the PELDOR/DEER results (the distance distribution and time trace of the variant 369/451 without cryo-protectant are shown in Fig. 6c (magenta line) and Supplementary Fig. 10).

To investigate whether the presence of the closed conformation of the apo protein seen for both variants was due to co-purified glutamine, we performed liquid chromatography mass spectrometry (LC-MS) experiments. Evaluation of the supernatant from purified variants showed that they did not contain detectable

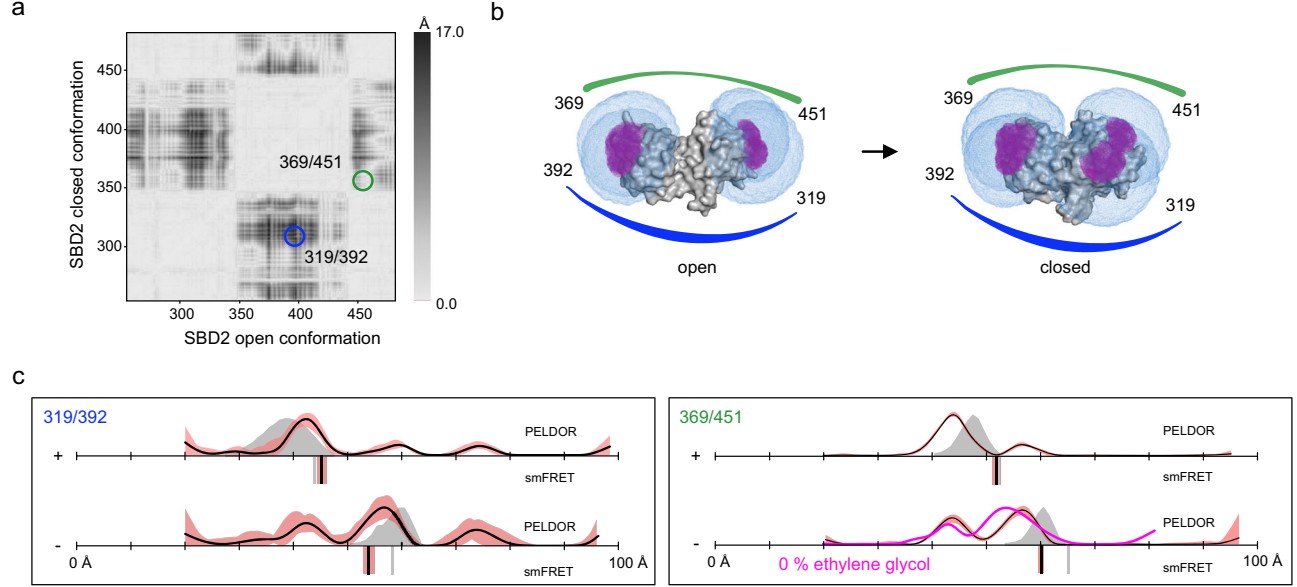

**Fig. 6 Distance measurements on SBD2. a** Difference distance map of SBD2 in the open (PDB-ID: 4KR5[34]) and closed (PDB-ID: 4KQP[34]) conformation. The dark spots mark protein regions that undergo large conformational changes relative to each other. The double variants for distance measurements are marked with circles. **b** Surface presentation of SBD2 (grey) in the open (left) and closed (right) conformation. The accessible volume of the spin label at four different labelling positions, calculated with mtsslWizard, are represented by magenta meshes and the accessible volume of FRET label maleimide-Alexa Fluor 647, calculated with FPS[67] is shown as blue meshes. The double variants that were used for experiments are identified with coloured lines, corresponding to the circles in (**a**). **c** Distance measurements with two different double variants of SBD2 without (−) and with (+ = 1 mM) substrate. The PELDOR/DEER results are shown above (grey curves for simulation, black curves for experiment) and the FRET distance below the axis (grey bars for simulation, black bars are the mean of $n = 3$ independent experiments). PELDOR/DEER results without cryoprotectant are shown as magenta curves. The red shade around the DEER data is the error margin calculated using the validation tool of DeerAnalysis[52]. The underlying principle of the validation tool is explained in the discussion section below. The red shade around the experimental smFRET distance are error bars based on the standard deviation of $n = 3$ independent experiments. Source data are provided as a Source Data file.

glutamine traces (μM concentrations would be needed to explain our observations, Supplementary Fig. 14).

In summary, for the SBD2 protein, both methods were able to discern the open- and closed state. But, the reason for the differences between the PELDOR/DEER and smFRET experiments remained elusive.

**Comparison 4: Type-III-secretion system effector protein YopO with a dynamic apo state**. PELDOR/DEER experiments are almost always conducted on bulk frozen samples and it is then not possible to study real-time molecular motions. However, dynamic processes can be studied indirectly by interpreting the width of distance distributions or by using freeze-quench experiments[81,82]. Recently, a combination of PELDOR/DEER and SAXS was used to investigate the conformational flexibility of the YopO protein from *Yersinia enterocolitica*[44] (Fig. 7a). The study revealed that the YopO/actin complex is a rigid entity with rather sharp PELDOR/DEER distance distributions, while the apo form of YopO adopts multiple conformations leading to a broader distance distribution (Fig. 7a, b, d). This is reflected in the PEL-DOR/DEER time trace for apo YopO where a sharp initial decay and a more damped oscillation were observed. These distinct features are due to the broad distribution of shorter inter-probe distances in the sample. By generating molecular models of the protein that simultaneously explained the PELDOR/DEER data and SEC-SAXS curves it was possible to obtain a coarse-grained insight into possible conformations of the apo protein in frozen solutions[44]. This approach could however not answer the question whether the generated structures are stable individual conformations of the apo protein or rather conformational states that each individual protein samples in a short period of time, i.e., on the sub-millisecond scale.

Such questions can be addressed by smFRET experiments. We hence labelled the YopO double variant 113/497 with the Alexa Fluor 555/647 dye combination and conducted smFRET measurements of YopO in the presence and absence of actin (Fig. 7c, d). Indeed, a clear shift from a high (with actin) to a low (without actin) FRET efficiency was observed. Interestingly, also the width of both FRET distributions was wider than expected (Fig. 7c), i.e., for YopO apo by 2.3-fold and 1.4-fold for holo (Supplementary Table 1). We next employed BVA[77] (Fig. 7e, f) to investigate, whether in contrast to HiSiaP (see above and Supplementary Fig. 5), sub-millisecond dynamics in apo YopO were responsible for this broadening. In BVA the shot-noise-limited standard deviation for a given mean FRET efficiency (STD of FRET, Fig. 7e, f) is compared against the experimental standard deviation within the burst (Fig. 7f). In accordance with PELDOR/DEER experiments (Fig. 7b, d[44]), smFRET suggests a dynamic apo state with sub-ms conformational dynamics and a static actin-bound conformation of YopO. This can be seen in the BVA plots where only deviations of the STD of FRET from the static line are seen for apo YopO (Fig. 7f) similar to a dynamic DNA hairpin structure and in contrast the static HiSiaP (Supplementary Fig. 5). These data underline the importance to identify FRET dynamics in bursts since only static distributions allow a meaningful interpretation of the mean FRET efficiency as a single distance in the macromolecular complex. With this knowledge we suggest that the average distances determined for the apo- and holo states of YopO by smFRET are as expected from the molecular models and show the same trends as the PELDOR/DEER data. In conclusion, we find a long distance in the holo state, which matches the simulated distance from the crystal structure, and an "apparent" shorter distance caused by multiple averaged conformations of YopO in the apo state (Fig. 7d). As previously noted, the PELDOR/DEER

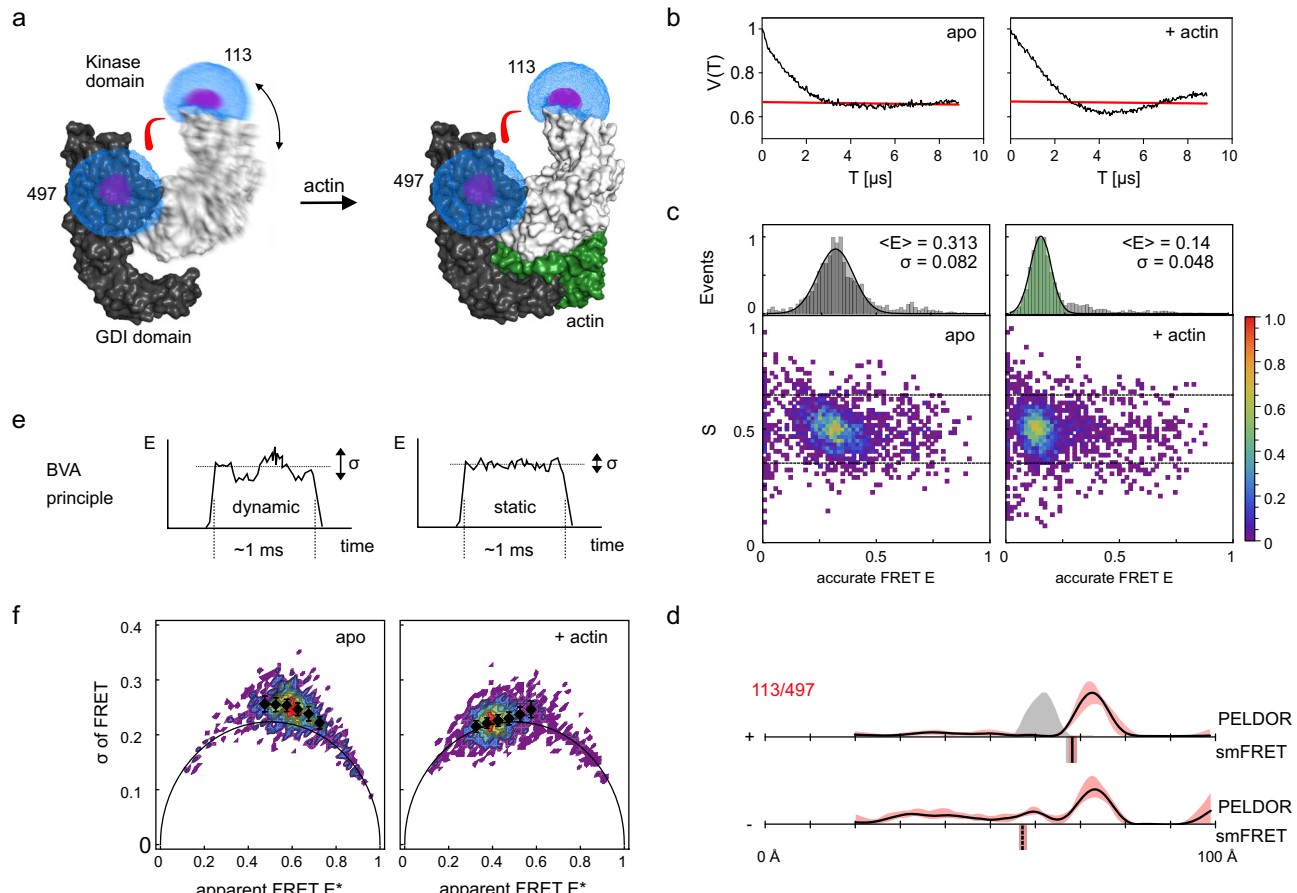

**Fig. 7 Distance measurements on dynamic system YopO. a** Model of the stabilization of YopO (kinase domain in grey, GDI domain in black) by binding of actin (green) (PDB-ID: 4CI6). The accessible volume of the spin label at two different labelling positions, calculated with mtsslWizard, are represented by magenta meshes and the accessible volume of FRET label maleimide-Alexa Fluor 647, calculated with FPS[67] is shown as blue meshes. **b** PELDOR time traces of spin labelled variant YopO$_{89-729}$ 113R1/497R1 without and with actin from our previously published study[44]. The red line indicates the background correction from DeerAnalysis[52] **c** smFRET data for Alexa Fluor 555 – Alexa Fluor 647 YopO$_{89-729}$ 113/497 in the presence and absence of actin. **d** Previously published[44] distance measurements with the YopO double variant without (−) and with (+) actin. The PELDOR/DEER results are shown above (grey curves for simulation, black curves for experiment) and the FRET distance below the axis (grey bars for simulation, black bars for experiment). Simulations for the apo state of YopO are not given, since no structural model for the apo state is available. The red shade around the DEER data is the error margin calculated using the validation tool of DeerAnalysis[52] (Reprinted from Structure, 27/9, ref. [44] with permission from Elsevier.). The underlying principle of the validation tool is explained in the discussion section below. The red shade around the experimental smFRET distance are error bars based on the standard deviation of $n = 3$ independent experiments. **e** Simplified schematic explaining the principle of the burst variance analysis (BVA): Dynamic systems show an increased variance in FRET efficiency during the measurement period (left) compared to the purely shot-noise limited variance of a static sample (right); please note that exchange of conformational states on timescales much faster than 100 μs can also give rise to an apparent static behaviour of the burst in BVA. **f** Burst variance analysis of the YopO measurement in (**c**) reveals a dynamic FRET state in the absence of actin (left) and a stabilized state in the presence of actin (right). The error bars represent the mean of $n = 3$ independent measurements and their standard deviation. Source data are provided as a Source Data file.

data[44] did not match the prediction from the crystal structure (Fig. 7d). This effect had been confirmed by additional PELDOR/DEER measurements and SAXS and could be explained by a slight re-orientation of the kinase domain[44]. It is possible that the longer linker of the smFRET labels masks this effect.

**Estimating the influence of linker length on the accuracy of predicted distance distributions**. While both PELDOR/DEER and smFRET can accurately measure inter-probe distances, one is ultimately interested in inter-residues distance between the labelled amino acids, which are different from the inter-probe distances. The common solution to interpret the distance data is therefore to build an in silico model of the labelled protein by placing the structure of the label (or a geometric model thereof) onto the molecular surface of the biomacromolecule and to

calculate its accessible volume[65–67,83]. Some programmes refine this accessible volume by considering preferred rotameric states of the label[65,83] or by selecting such conformations that are close to the molecular surface[67,71]. Unfortunately, it is difficult to build a model that accurately reflects the rotameric state of the labels, their interactions with the protein surface or solvent, and the molecular motion of the protein backbone. Hence, a major part of the apparent distance inaccuracy (i.e., the mismatch between predicted and experimental distance) of PELDOR/DEER and smFRET is actually caused by the in silico models[65–67,83]. Nevertheless, many examples in the literature have shown that PELDOR/DEER distance distributions can be predicted rather well (±2–4 Å) by determining all possible distances between the modelled spin centres in such ensembles[71,84,85]. For smFRET, it was shown that the average distance can be predicted with an accuracy of ±5 Å[76]. Calculating distance distributions is more

difficult for smFRET due to the indirect way of distance determination via FRET efficiency[67,86,87], unless lifetime-based approaches are used[78]. The resulting averaging effects between simulated distance distributions and FRET-averaged measured distances are discussed in detail in ref. [86].

No matter which method is used, the prediction accuracy in the end depends on the correctness of the modelled label. Considering the topic of this work, we asked ourselves, how the prediction uncertainty is influenced by geometrical factors, i.e., the distinct linker length of typical spin- and fluorescence labels (Fig. 2) and protein-label interactions (Fig. 8).

For the following simulations, we used a slightly modified version of mtsslWizard[71]. The programme was used to attach in silico models of two different labels (MTSSL and Alexa Fluor 647) to the open-state HiSiaP protein (PDB-ID: 2CEY) and to generate ensembles of rotamers at positions 55, 58, 112, 134, 175, and 225. To simulate the effect of protein-label interactions, we replaced a certain percentage of randomly selected rotamers in each ensemble by one randomly chosen rotamer. In our simulation, these "sticky" rotamers represented an immobilized label.

We arbitrarily defined a weak interaction to lead to 10% of the rotamers in a given ensemble to occupy the same position, while the remaining 90% of rotamers were randomly distributed in the accessible volume. The simulation was run for immobilized/mobile ratios of 10/90, 50/50, and 100/0 (Fig. 8). We then determined the average distances between each pair of the "sticky" ensembles (this simulates a set of distance measurements), as well as the average distances between the corresponding pairs of the original ensembles (i.e., without "sticky" rotamers, which corresponds to the prediction of a particular distance by the accessible volume approach) and calculated the prediction errors. The whole procedure was repeated 1000 times to achieve a statistical distribution of the "interaction site" within the ensembles. The results are summarized in Fig. 8b: For a weakly immobilized label (Fig. 8b, magenta), the prediction error was low, for both MTSSL (width of error histogram: $\sigma = 0.4$ Å) and for Alexa Fluor 647 ($\sigma = 0.6$ Å) (Fig. 2). This changed markedly when the label interacted more strongly with the protein surface. If the label occupied a fixed position half of the time, the prediction error increased considerably (Fig. 8b, beige). Whereas the error was still relatively small ($\sigma = 1.8$ Å) for MTSSL, larger errors of up to $\pm 10$ Å ($\sigma = 3.2$ Å) quite frequently occurred for Alexa Fluor 647. If the interaction was even stronger, i.e., for a completely immobilized label (100%, blue), it was quite likely to observe a very high prediction error for MTSSL ($\sigma = 3.8$ Å) but especially for Alexa Fluor 647 ($\sigma = 6.6$ Å). Note that our approach neglects movements of the protein backbone, which will further increase the observed errors. An example for such an immobilized label is the crystal structure of the matrix metalloproteinases MMP-12 in complex with a fluorophore labelled Cy5.5 inhibitor[88] (Fig. 8b), which has a chemical structure related to Alexa Fluor 647 (Fig. 2). In the complex, a large part of the label is bound to the surface of the protein with its sulfonic acid groups interacting with the positive charges of lysine and arginine residues. Figure 8b shows the size of the accessible volume for the label (blue) for comparison. Similar observations have been made for the MTSSL label, for instance in the case of the Spa15 chaperone[80].

## Discussion

We performed PELDOR/DEER and smFRET experiments on three substrate binding proteins (HiSiaP, MalE, SBD2) and the YopO protein to conduct a comprehensive cross-validation of the two methods. One of our goals was to determine the distance accuracy by comparison of simulated mean inter-probe distances

vs. experimental ones and to see if both techniques correctly monitor ligand-induced structural changes. For this purpose, we used the same labelling sites for both methods and measured the inter-probe distances in the presence or absence of the respective ligands. Both methods showed a good consistency with each other and towards structural models. For a quantitative comparison, we selected 15 datasets that were measured in this study and where a monomodal distance distribution was observed. We computed the difference between the average experimental and simulated distances for both PELDOR/DEER and smFRET and plotted the data in Fig. 8c, d. The PELDOR/DEER and smFRET measurements on the same double variant differed by about 5 Å with an overall spread of $\pm 10$ Å (Fig. 8e). The spread of the datapoints agrees well with the error margins of $\pm 3.5$ and $\pm 5$ Å that are commonly given for PELDOR/DEER and smFRET, respectively (grey shades in Fig. 8c, d)[65–67,83,89]. Within the dataset, no systematic offset was observed between PELDOR/DEER or smFRET. However, Fig. 8c indicates a trend, where longer distances show a larger deviation in the prediction accuracy for both methods.

To an extent, the above-described differences between the PELDOR/DEER and smFRET results will be related to the different nature of the labels and in particular to their differing linker lengths (compare Fig. 2). Figure 8e shows that for most of our measurements, the difference between the two methods is larger than can be explained by the different linker lengths for a freely rotating label (otherwise, the black and white data points in Fig. 8e should coincide). The simulations in Fig. 8a, b reveal that already moderate protein-label interactions can lead to distance measurements that are seemingly inexplicable. This is vividly reflected in our first example (Figs. 3, 4). Luckily, there are experimental approaches to detect strong protein label interactions in smFRET (Fig. 4) as well as for PELDOR/DEER. For the latter, the shape of room temperature cw spectra and abnormally shaped Pake patterns provide hints towards strongly immobilized spin labels[90,91]. Depending on the label, its degree of immobilization, and especially at high magnetic fields, it can be necessary to consider orientation selection effects by collecting multiple PELDOR/DEER time traces at different frequency offsets[91–93].

Based on the above, three seemingly obvious solutions emerge to improve the distance accuracies of both techniques:

(i) The use of probes with shorter linkers. It is challenging to shorten the length of e.g., the MTSSL label any further (Fig. 2), and short inflexible linkers such as the two-armed Rx spin label[94] are more likely to disturb or artificially stiffen the protein structure. Alternative EPR labels such as Gd-, Cu- or trityl labels with different linker types and lengths are under active research[95–97]. For smFRET labels, shorter linkers are not only a challenge of chemical synthesis, but also for the implementation of the method itself, because the free rotation of the dye is the basic requirement for distance simulations based on the accessible volume and to exclude orientational effects on FRET efficiency[67]. This requirement would be hard to meet with shorter and thus also more rigid linkers.

(ii) The prevention of label-protein interactions. A systematic reduction of label-protein interactions is challenging and its feasibility depends on the type of macromolecule. For nucleic acids for instance, the molecular surface will be predominantly negatively charged. Still, many parameters such as the counter-ion concentration and the fold of the nucleic acid (especially for RNA) can strongly impact label-nucleic acid interactions. For proteins, it is even more difficult to predict how the label will interact with the macromolecular surface and unexpected results are more likely. Nevertheless, we showed for HiSiaP that switching

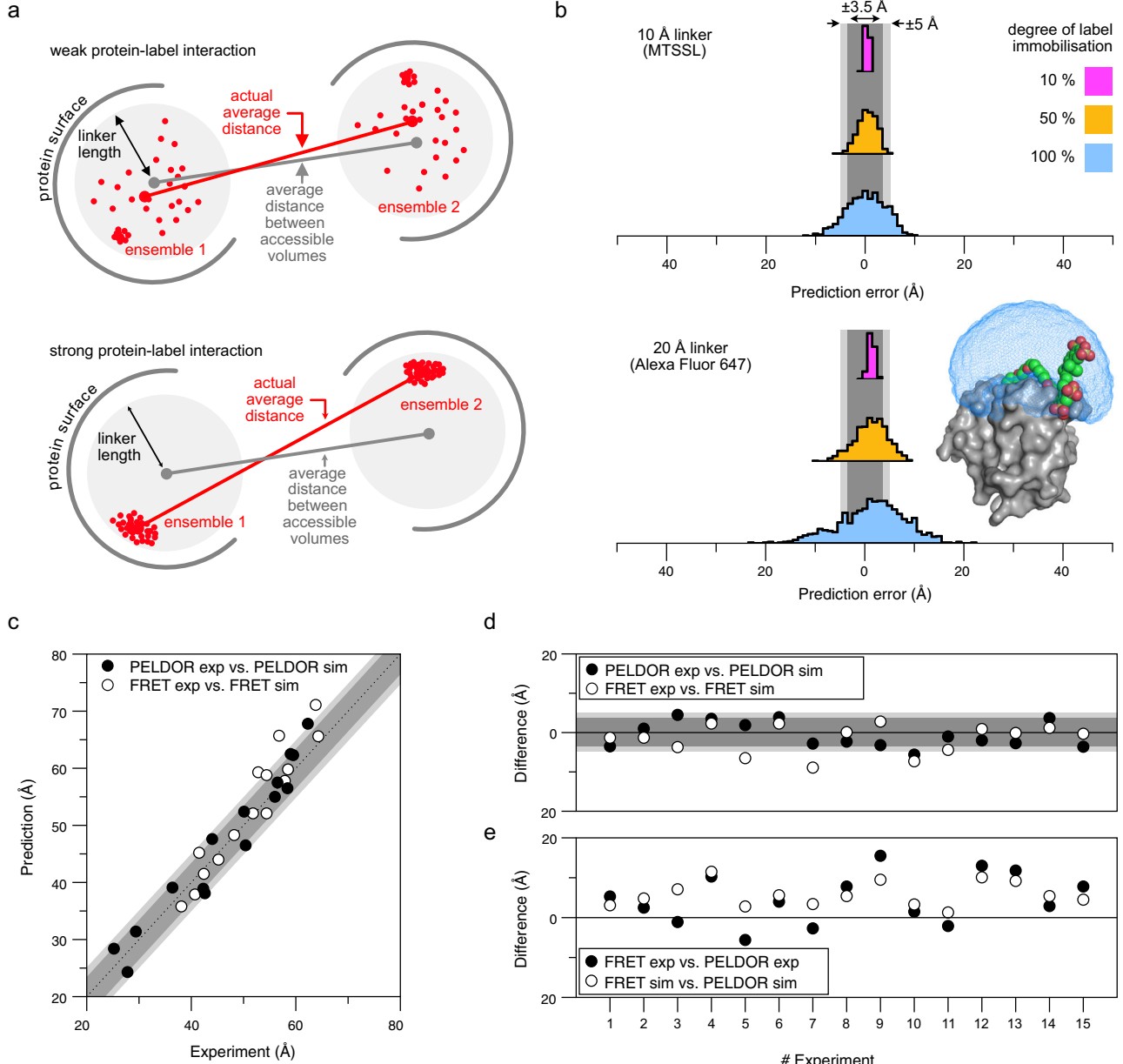

**Fig. 8 Comparison of PELDOR/DEER and smFRET measurements and the influence of linker length on the correlation between experimental and predicted distances. a** Multiple ensembles of spin- and fluorescence labels were simulated with mtsslWizard using the open form of HiSiaP (PDB-ID: 2CEY[29]) and the labelling sites 55, 58, 112, 134, 175, and 225. In the schematic, the radius of the sphere represents the length of the linker that connects the fluorophore or spin centre to the C-alpha atom of the labelled residue. Interactions with the protein surface (grey arcs) are indicated and lead to a clustering of labels at that position. Depending on the degree of interaction between protein and label, the accessible volume approach becomes less accurate. **b** Histograms of 1000 simulations described in (**a**) with a 10 Å linker (upper plot) and 20 Å linker (lower plot) and varying degree of protein label interaction. The percentage indicates how many percent of the 1000 dummy atoms are localized at the interaction site. As an example for a long (20 Å) and immobilized linker, the protein structure of MMP-12 (matrix metalloproteinases, PDB-ID: 5L79[88]) in conjugation with a Cy5.5 fluorophore (K241, coloured spheres) was selected. The surface of the protein is shown in grey and the accessible volume of the fluorophore, calculated with FPS[67] is shown as a blue mesh. The dark- and light-grey shades represent the error margins of ±3.5 and ±5.0 Å that are often given for PELDOR/DEER and smFRET experiments, respectively. **c** Predicted vs experimental smFRET or PELDOR/DEER distances of datasets that were measured with both methods in this study. **d** As (**c**) but the differences are plotted against the experiment number in Supplementary Table 2. **e** Comparison of the distances determined by PELDOR/DEER or smFRET and the simulation for the same experiment. Source data are provided as a Source Data file.

fluorophores to alter charge and linker length allowed us to circumvent this problem and to detect the expected conformational change and distances (Fig. 4).

(iii) Improving in silico labelling approaches. So far, even time-consuming molecular dynamics approaches have not been shown to be much more accurate than the accessible volume approach in benchmarks studies[71,84,85]. It should

be noted that new and promising approaches to tackle this problem are constantly developed[98,99]. Still, no matter how sophisticated the prediction algorithm, it will be difficult to obtain absolute certainty.

A good, yet cumbersome option is to measure as many distances as possible and then to investigate, whether a particular

observation is truly independent of the labelling site and label. Cross-validation of the results with a second (preferably label-free) method such as SAXS[44,100] is another option that is however not feasible for every system (membrane proteins in detergent or lipid belts are for instance not an easy target for SAXS[101]).

Some PELDOR/DEER-specific considerations emerge from our results. The use of cryo-protectants had a significant impact on our distance measurements on MalE. These substances are routinely added to prevent protein aggregation during freezing and thereby to improve the phase memory time of the samples. At the high concentrations that are typically used (10–50% v/v glycerol or ethylene-glycol), the small molecules might interact with the protein and induce a different conformation of the spin label or the protein itself[102,103]. Note that the proteins in our study (such as MalE) might be particularly prone to such problems, because they have deep surface crevices that can easily bind such small molecules. We did not see a significant influence of the cryo-protectant for the SBD2 example and the same was true for previous measurements on VcSiaP (a close homolog of HiSiaP from example 1) (Supplementary Fig. 15). Nevertheless, especially when unexpected results are found, a control measurement without cryo-protectant or a different cryo-protectant from the large arsenal of such substances should be performed[104]. Efforts to develop experimental procedures that allow to reduce the amount of cryo-protectant or to completely avoid their addition, for example by rapid freeze quenching are a promising route to circumvent this problem[102,105]. Ultimately, it is desirable to perform the PELDOR/DEER measurements at physiological temperatures. Conformational equilibria are temperature dependent and are thus affected by the freezing procedure, which, despite all efforts, is still slow compared to the time scale of most molecular motions. The quantitative differences between PELDOR/DEER and smFRET that were observed in the case of SBD2 might be caused by the different experimental temperatures. Unfortunately, room temperature PELDOR/DEER measurements on proteins in solution are very challenging and it is, therefore, no simple task to test this. Newly developed labels, such as the trityl spin labels are an important step on the road towards room temperature PELDOR/DEER experiments on proteins in solution[106,107]. Another factor that might contribute to the observed deviations between PELDOR/DEER and smFRET, is the sample concentration. Whereas smFRET experiments are conducted in extremely dilute solutions, standard Q-band PELDOR/DEER experiments are performed at micromolar to nanomolar concentrations[108].

It is a strong point of PELDOR/DEER that distance distributions rather than average distances can be readily obtained. However, it is important to remember that the shape of these distributions depends on a number of parameters such as the quality and especially the length of the underlying PELDOR/DEER time trace[45,109]. Unfortunately, its maximum length cannot be arbitrarily chosen, since the refocused echo signal quickly decreases with an increasing time window (Fig. 1d). Hence, a tradeoff between signal strength and length of the time-window must be made for each sample, where it is usually more important to have a longer time window than a very high SNR. The conversion of time traces into distance distributions is often solved by a two-step analysis of first fitting and removing the intermolecular background (Fig. 1d bottom, dashed red line) and then applying Tikhonov regularization to extract the distance information[51,52]. The procedure introduces a regularization parameter $\alpha$ that describes a compromise between the smoothness of the distance distribution and how well it reproduces the experimental time-trace[51]. Because the true shape of the distribution is of course unknown, this procedure inevitably introduces a degree of uncertainty. The two-step procedure works well

for high-quality data, where more than a complete oscillation period of the signal was recorded (e.g., Supplementary Fig. 2). In practice, this is not always the case and the separation of the intermolecular background becomes a source of uncertainty. The evaluation feature of the DeerAnalysis software can be used to visualize the impact of this problem on the distance distribution. The software systematically varies parameters, such as the starting time of the background fit to obtain a mean value, a standard deviation, as well as upper- and lower limits for each point of the distance distribution[110]. In our comparisons above, the red shade around the distance distribution was produced with this feature. Recently, new data processing algorithms have been developed that for example calculate the distance distribution in a more robust one-step analysis and also consider the noise level in the raw data to estimate the uncertainty of the distance distribution[111,112]. Yet another approach is DeerNet[53,113], an artificial neural network that was trained on a large database of simulated data. This latter method is independent of user-adjustable parameters. As a comparison, we processed the key datasets measured in this study with DeerNet and reassuringly found very similar results Supplementary Figs. 19, 20.

In a recent study, aliquots of the same PELDOR/DEER samples were analyzed by seven EPR laboratories[114]. While the resulting distance distributions were overall quite similar, the variation between the individual labs was not fully covered by the error margins calculated with the different processing algorithms. It was concluded that this was caused by the uncertainty of background separation due to the different lengths of the time traces that were recorded by the different labs and an overlap of the excitation bands of observer- and pump pulses which is not yet accounted for by the algorithms[111,114].

A major obstacle for the determination of inter-probe distances via smFRET is the conversion of the experimentally derived setup-dependent apparent FRET efficiency E* values first into (i) accurate FRET E values (Supplementary Fig. 16) and (ii) inter-probe distances. We provide a full overview of all parameters required for distance determination via smFRET in Supplementary Table 3 for different protein variants. This correction is not required for lifetime-based approaches, which allow to obtain distance distributions from the lifetime decays directly[78]. However, a quite common procedure is the ratiometric determination of FRET efficiency with subsequent conversion into accurate FRET E and later into distances[16,89,115]. We will here discuss those steps that are most problematic. The correction for systematic errors introduced by the apparatus (background, spectral crosstalk, and differences in donor/acceptor detection and fluorescence quantum efficiency; Supplementary Fig. 16) is directly carried over to the corrected FRET efficiency $E$ values (step i). Later, in step (ii) Förster-radius determination is required for the conversion of accurate E values into distances.

The introduced systematic errors in step (i) are largely dominated by the $\gamma$ correction factor describing differences in acceptor-to-donor detection- and fluorescence-quantum efficiency. Obtaining a reliable $\gamma$ factors is thus particularly important, also because cyanine-based donor and acceptor dyes, which are used in our manuscript, show large changes of fluorescence quantum yield depending on their specific environment[116–119]. The correction step, however, requires multiple experiments with the same pair of labels (if possible, in the same environment), but with distinct FRET efficiencies, e.g., obtained via a conformational change. Such local $\gamma$ values are, however, often not accessible and fluorophores often experience changes of their fluorescence quantum yield impacting $\gamma$. In our case, ligand-induced conformational changes provided access to two FRET efficiency states with (often) identical fluorophore properties (Figs. 3–6). This assumption is, however, not always valid as was

seen in smFRET experiments of HiSiaP for distinct conformational states (Figs. 3, 4). In cases where only a single FRET efficiency state is available for each label position, a global $\gamma$ factor has to be used, e.g., derived from the same combination of labels at different label-positions on the target, using the crude assumption that the fluorophore environment remains identical.

Another key problem related to step (ii) of the analysis procedure is the determination of the Förster radius $R_0$. Since the same fluorophore may show different fluorescence quantum yields, spectra etc. subject to the its biochemical environment, $R_0$ will vary in the range of $+/-0.3$ nm (impacting the determined distances by 0.3–0.5 nm). Consequently, accurate calculation of $R_0$ can improve the distance determination significantly[89], yet it requires information on the donor fluorescence spectrum, donor quantum yield, acceptor absorption spectrum, the refractive index of the donor–acceptor intervening medium and their relative orientation. Problematically, many approaches currently in use for their determination overlook heterogeneity, particularly when stochastic labelling approaches are used. Particularly challenging is also the determination of the refractive index and the dye orientation factor $\kappa^2$. The refractive index is often simply estimated to be an average value of $n \approx 1.4$ considering the values for water ($n = 1.33$) and that of proteins ($n = 1.5$)[120]. $\kappa^2$ on the other hand is often idealized by the idea of non-interacting fluorophores that undergo free rotation during the donor fluorescence lifetime to a value of ~2/3. The distance-error resulting from an incorrect $\kappa^2$ can be estimated, and studies showed the effects $\kappa^2$ values can have on the FRET-derived distance estimation[121]. As a rule of thumb fluorescence anisotropy of the individual dyes, e.g., measured via steady-state anisotropy should be less than 0.2. This approximation, however, does not hold for large organic dyes or dyes with short lifetimes, since rotational correlation times (determined indirectly from fluorescence anisotropy decays) will be similar to the fluorescence lifetime.

This is the case for Alexa Fluor 555 as donor dye and prohibits its proper use for lifetime-based experiments. A concern might be that the donor dye used here (Alexa Fluor 555) has varying $R_0$-values and with that a poor distance accuracy. Based on this, the question might arise, why Alexa Fluor 555 was chosen as a label for this study. Our goal was to provide a comparison of smFRET and PELDOR/DEER using the most commonly used labels. By doing so we will allow other users to adapt the analysis routines, e.g., for obtaining accurate FRET values and distances. Notably, the most commonly used dye for smFRET studies is Alexa Fluor 555. Furthermore, our study shows a very good agreement between smFRET and PELDOR/DEER in both a qualitative (trends of distance changes) and quantitative sense (distance accuracy in comparison to AV calculations for smFRET or rotamer libraries for PELDOR/DEER). In the instances where discrepancies were observed between the techniques (or simulations), we managed to identify the reason for it. To validate the idea that Alexa Fluor 555 is as useful as other dyes for quantitative smFRET, we compared smFRET-derived distances of Alexa Fluor 555 (a cyanine) to Alexa Fluor 532 and 546 (both rhodamine dyes). Using two sets of cysteine pairs in both MalE and SBD2 we derived simulated and calculated distances as shown in Supplementary Fig. 21 and Supplementary Table 4. The data and a plot of $\Delta R$-values reveals that the deviations (theoretical vs. experimental interprobe distance) are indeed very similar for all dye pairs. Importantly, we can see that certain labelling positions show larger deviations due to sticking of the fluorophores, which also seems independent on the choice of dye.

In summary, the distance variations seen in smFRET are fairly consistent with an error assessment described by Hellenkamp et al.[89]. Since all distances we studied here are in the range of $0.8 * R_0$ to $1.3 * R_0$, we assume that errors associated to

background and spectral cross-talk ($\alpha$, and $\delta$ errors play a minor role, i.e., $\Delta R < 1$ Å) are minor as suggested Hellenkamp et al. Therefore, the major contribution is based on wrong $R_0$ determination or incorrect $\gamma$-values (as discussed above). Based on this we can estimate $\Delta R_\gamma \approx 1$ Å and $\Delta R_{R0} \approx 3$–4.5 Å (depending on the distance) based on a relative error in $\gamma$ and $R_0$ of 10 and 7%, respectively, which is in full agreement with the accuracy found in our study (Fig. 8 and Supplementary Fig. 21).

Ironically, structural dynamics are usually the reason why PELDOR/DEER and smFRET are applied in the first place, but at the same time, these processes heavily impair the possibility to interpret and use inter-probe distances (or distance distributions) in a straightforward manner. In essence, this is because (time-) averaged distances cannot be interpreted easily in light of static structures. Thus, the use of inter-probe distances in structural modelling requires a verification of the degree of structural dynamics. Since smFRET experiments are performed with individual molecules in liquid solution at room temperature, it allows to characterize fast structural dynamics over a range of different timescales ranging from nano- to milliseconds[20,122]. In contrast, PELDOR/DEER experiments are almost always conducted on frozen bulk samples and it is not possible to directly study dynamics in real-time. Yet, PELDOR/DEER data can be readily converted to distance distributions and the shape of the latter contains information about conformational heterogeneity of the sample (see above for important caveats on interpreting the shape of distance distributions). The YopO example demonstrates this difference between the bulk-point of view of PELDOR/DEER and the single-molecule point of view of smFRET. From the broad distance distributions of the original PELDOR/DEER data, it was correctly inferred that the YopO protein is flexible and adopts different conformational states in its apo state[44]. However, it was not possible to differentiate between static or dynamic nature of the conformational heterogeneity in the sample. With smFRET investigations it became clear that apo YopO is indeed dynamic and shows sub-millisecond conformational motion that depend on the presence or absence of actin (Fig. 7). Recently, we studied YopO via multi-parameter photon-by-photon hidden Markov modelling, where the underlying apo distribution of YopO could be resolved[123]. We showed that apo YopO consists of two major FRET peaks that average on a sub 100 μs timescale to give one apparently "static" population. The addition of the ligand stabilized one of the two conformers (low FRET).

Both PELDOR/DEER and smFRET are valuable tools for integrative structural biology. Overall, we found a reasonable agreement of the determined distances with an average ±5 Å spread between the two methods. But, our experiments also revealed discrepancies that might have led to wrong interpretations if only one of the methods had been used. We could show that these differences were partly due to the distinct labels and label-protein interactions. Thus, more reliable methods to predict or prevent label-protein interactions are urgently needed. Also, the much longer linkers used for smFRET can be problematic. For PELDOR/DEER, the use of cryogenic temperatures and cryoprotectants was shown to influence conformational changes of the MalE protein. Our study clearly shows that a combination of PELDOR/DEER and smFRET provides highly complementary and synergistic insights into the conformational states of macromolecules. Hence, the development of spectrometers and microscopes, as well as standardized data processing approaches[20,122], which can also be used by non-experts, would be very beneficial to structural biology[124].

## Methods

**Selection of labelling sites.** Dependent on the particular method (PELDOR/DEER or smFRET) for which the protein variants were produced, we used different software to calculate suitable labelling positions. For spin label positions we used

mtsslSuite (www.mtsslsuite.isb.ukbonn.de) and calculated a difference distance map between the open and closed conformation (as shown in results)[125]. With this map we identified regions with large conformational changes and selected amino acids inside these regions, which are located on the surface of the protein to obtain a good accessibility. For smFRET studies, residues were rated based on different parameters such as solvent exposure or conservation to obtain a labelling feasibility estimate. Residues were selected that showed large distance changes between apo and holo (or no distance change as negative control).

**Expression and purification of HiSiaP and VcSiaP.** The HiSiaP and VcSiaP variants were expressed and purified according to an established protocol[73]: To prevent co-purification of sialic acid, an overnight culture of *E. coli* C43 cells in LB medium was pelleted by centrifugation and washed twice with M9 minimal medium, supplemented with 5% glycerol. Six liters of the M9 medium (including 5% glycerol, 100 µg/ml ampicillin, 2 mM $MgSO_4$, and 0.1 mM $CaCl_2$) were inoculated with 10 mL of the washed cells ($OD_{600}$ = 5.0–6.0) and incubated at 37 °C for 14–16 h with shaking until an $OD_{600}$ of 0.6–1.0 was reached. Protein expression was induced with 500 mg/l L(+)-arabinose and allowed to proceed for 5 h at 37 °C. For purification, the resuspended cells (50 mM Tris-HCl, pH 8, 50 mM NaCl, 10% glycerol) were lysed with a cell disrupter (Constant System, Daventry, Northamptonshire, UK), and centrifuged to remove cell debris. The supernatant was loaded onto a benchtop $Ni^{2+}$ NTA column (GE Healthcare, equilibrated in buffer A: 50 mM Tris-Cl, pH 8, 50 mM NaCl), washed with 100 mL of buffer A and eluted by supplementing buffer A with 500 mM imidazole. This step was followed by an ion exchange chromatography (Biorad ENrichQ 10/100) using buffer A and a linear gradient from 0.05 to 1 M NaCl. Size exclusion chromatography was used to remove remaining impurities. A Superdex 200 16/600 column (GE Healthcare) equilibrated with buffer A was used for this step. Throughout the procedure, buffers were supplemented with 1 mM Tris(2-carboxyethyl)phosphine (TCEP) to avoid dimerization of the cysteine variants and the purity was checked after each step with SDS-PAGE. The purified protein solution was concentrated to 20 mg/mL and stored at −80 °C until labelling.

**Expression and purification of MalE and SBD2.** The MalE double-cysteine variants were produced by PCR site-directed mutagenesis, using the T7/lac bacterial expression vector pET-23b(+). *E. coli* BL21(DE3)pLysS competent cells containing the plasmids of the MalE variants[75] were grown at 37 °C in LB medium supplemented with ampicillin (0.1 mg/ml) and chloramphenicol (0.05 mg/ml) until an optical density ($OD_{600}$) of 0.6–0.8 was reached. Protein overexpression was induced by addition of 0.25 mM isopropyl β-D-1-thiogalactopyranoside (IPTG). The cells were harvested after 2 h and resuspended in resuspension buffer (50 mM Tris-HCl, pH 8.0, 1 M KCl, 10% glycerol, 10 mM imidazole, and 1 mM dithiothreitol). DNase 500 ug/ml (Merck) and 0.2 mM phenylmethylsulfonyl fluoride (PMSF) were added before the cells were lysed using an Ultrasonic homogenizer (Branson Digital Sonifier; 25% amplitude, 0.5 s duration of "on" and "off" ultrasound pulses) equipped with a micro-tip probe of 3 mm diameter. Cell debris were removed by centrifugation at 5000 × $g$ for 30 min at 4 °C, followed by a further ultracentrifugation step at 208,400 × $g$ for 1 h at 4 °C. The MalE protein variants were purified by affinity chromatography using the $Ni^{2+}$-sepharose resin (GE Healthcare), which was pre-equilibrated with 10 CV resuspension buffer. Then, the cell lysate supernatant was loaded onto $Ni^{2+}$-sepharose resin. After two subsequent washing steps with 10 CV resuspension buffer and further 10 CV wash buffer (50 mM Tris-HCl, pH 8.0, 50 mM KCl, 10% glycerol, 30 mM imidazole and 1 mM DTT) the target protein was eluted with 2.5 CV elution buffer (50 mM Tris-HCl, pH 8.0, 50 mM KCl, 10% glycerol, 250 mM imidazole and 1 mM DTT). Protein containing fractions were pooled and dialyzed overnight against 100 volumes of dialysis buffer (50 mM Tris-HCl, pH 8.0, 50 mM KCl and 10 mM DTT) to eliminate the excess of imidazole. Next, the dialysis buffer was further exchanged with additional 100 volumes of storage buffer (50 mM Tris-HCl, pH 8.0, 50 mM KCl, 50% glycerol, and 10 mM DTT). Protein aliquots were stored at −20 °C until required.

The purification of SBD2 was adapted from Gouridis et al.[33]. All the overexpression and purification steps were conducted as described for MalE variants, except that the SBD2 double-cysteine variants were produced using the araBAD bacterial expression vector pBAD/His. Therefore, the protein overexpression was here induced by the addition of 0.2% L-arabinose.

**Expression and purification of YopO.** YopO C219A L113C/L497C was produced according to an established protocol[44]. An overnight culture of *E. coli* Rosetta (DE3) cells containing the expression plasmid was used to inoculate 1 l of LB medium containing 100 µg/ml ampicillin and 37 µg/ml chloramphenicol. At an $OD_{600}$ of ~0.8–1.0, the culture was induced with 0.1 mM IPTG (isopropyl β-D-1-thiogalactopyranoside) and the protein was expressed overnight with shaking at 16 °C. The cells were then harvested by centrifugation with 4000 × $g$ at 4 °C for 20 min. The cell pellet was resuspended in 5 times w/v of lysis buffer (50 mM Tris-Cl pH 8.0, 300 mM NaCl, and 3 mM DTT). The cells were lysed by sonication and cell debris was removed by centrifugation for 20 min at 4 °C with 50.000 × $g$. The supernatant was incubated with GST sepharose (GE Healthcare, equilibrated with lysis buffer), for 1 h at room temperature. After washing with 50 ml wash buffer

(50 mM Tris-Cl pH 8.0, 150 mM NaCl and 3 mM DTT) the resin was incubated with 20 ml of PBS pH 8.0, 1 mM DTT, 1 mM EDTA and 0.1 mg PreScission protease (GE Healthcare) overnight at 4 °C. The flowthrough of this step was collected and diluted with 50 ml ion-exchange buffer A (50 mM Tris-HCl pH 8.0) and 90 µl of 2 M DTT. The protein was loaded onto a MonoQ column (GE Healthcare) using ion-exchange buffer A (50 mM Tris-HCl pH 8.0) and eluted with a gradient of ion-exchange buffer B (50 mM Tris-HCl pH 8.0, 1 M NaCl). To remove further impurities, a size-exclusion chromatography was performed with buffer containing 50 mM Tris pH 8, 50 mM NaCl and 1 mM DTT. Fractions containing pure protein were pooled and used for spin labelling experiments.

**LC-MS.** The LC-MS analysis was performed on an HTC esquire (Bruker Daltonic) in combination with an Agilent 1100 Series HPLC system (Agilent Technologies). Analysis gradient: 5 → 100% MeCN (solvent B)/0.1% formic acid (solvent A) in 20 min at a flow rate of 0.4 mL $min^{-1}$ using a Zorbax Narrow Bore (2.1 × 50 mm, 5 µm) C18 column (Agilent Technologies).

**Spin labelling.** In the first step, the reducing agents in the protein solution were removed with a PD10 desalting column (GE Healthcare), using a buffer based on 50 mM Tris, 50 mM NaCl, pH 8 without TCEP or DTT. Immediately after elution from the column, the protein eluate was treated with 5 times excess per cysteine of the nitroxide spin label MTSSL (Toronto Research Chemicals, Canada), dissolved in DMSO. The labelling was carried out for one hour at room temperature under gentle shaking. Afterwards, the protein was concentrated and another PD10 desalting column was used to remove free spin label. The protein eluate was again concentrated to ~20 mg/mL. The labelling was verified and quantified with continuous-wavelength EPR spectroscopy (cw-EPR)[90] using an EMXnano X-band EPR spectrometer from Buker (Billerica, MA). The spin labelled proteins were diluted to a concentration of 25 µM with standard buffer and a total volume of 10 µL sample was prepared into a glass capillary, sealed with superglue. The magnetic field of the cw-EPR spectrometer at room temperature were set to a centre field of 3448 G and the microwave frequency to 9.631694 GHz. The microwave power was set to 2.5 mW, the power attenuation to 16 dB and the receiver gain to 68 dB. The cw-EPR spectra were recorded with a sweep width of 150 G, a sweep time of 10.03 s with 20.48 ms time constant, and 1 G modulation amplitude. For every sample 350 cw-EPR spectra were averaged to obtain a good SNR. The concentration of the spin label and the labelling efficiency were determined with the Bruker software Xenon by double integration of the cw-EPR spectrum.

**Fluorophore labelling.** Proteins were labelled according to an established protocol[33,74]. The cysteines were stochastically labelled with the maleimide derivative of the dyes TMR, Alexa Fluor 555, Alexa Fluor 647 and Cy5 (ThermoFisher Scientific). His-tagged proteins were incubated in 1 mM DTT to keep all cysteine residues in a reduced state and subsequently immobilized on a Ni Sepharose 6 Fast Flow resin (GE Healthcare). The resin was incubated 2–4 h at 4 °C with 25 nmol of each fluorophore dissolved in 1 ml of labelling buffer 1 (50 mM Tris-HCl pH 7.4, 50 mM KCl, 5% glycerol) and subsequently washed sequentially with 3 ml labelling buffer 1 and buffer 2 (50 mM Tris-HCl pH 7.4, 150 mM KCl, 50% glycerol) to remove unbound fluorophores. Bound proteins were eluted with 500 ml of elution buffer (50 mM Tris-HCl pH 8, 50 mM KCl, 5% glycerol, 500 mM imidazole) The labelled protein was further purified by size-exclusion chromatography (ÄKTA pure, Superdex 75 Increase 10/300 GL, GE Healthcare) to eliminate remaining fluorophores and remove soluble aggregates. For all proteins, labelling efficiencies were higher than 70%, and donor–acceptor pairing at least 20%.

**PELDOR/DEER spectroscopy.** If not indicated in the results, the standard PELDOR/DEER samples were prepared and the measurements were set up as described in the following. The proteins and additives were mixed and diluted to a concentration of 15 µM in a volume of 40 µL with PELDOR/DEER buffer (100 mM TES pH 7.5, 100 mM NaCl in $D_2O$). The substrate concentrations were 1 mM N-acetyl neuraminic acid for HiSiaP, 1 mM maltose for MalE and 100 µM glutamine for SBD2. The solutions were supplied with 40 µL deuterated ethylene glycol, transferred into a 3 mm quartz Q-band EPR tube, and immediately flash-frozen and stored in liquid nitrogen.

The PELDOR/DEER experiments were measured on an ELEXSYS E580 pulsed spectrometer from Bruker in combination with an ER 5106QT-2 Q-band resonator using the XEPR software. The temperature was set to 50 K with a continuous flow helium cryostat (CF935, Oxford Instruments) and a temperature control system (ITC 502, Oxford Instruments). The PELDOR/DEER time traces were recorded with the pulse sequence π/2($v_A$)-$\tau_1$-π($v_A$) − ($\tau_1$ + t) − π($v_B$)-($\tau_2$-t)- π($v_A$)- $\tau_2$-echo. The frequency $v_A$ of the detection pulses were set 80 MHz lower than the frequency of the pump pulse $v_B$, which was set to the resonator frequency and the maximum of the nitroxide spectrum. Typically, the shot repetition time was 1000 µs and the lengths of $\tau_1$ and $\tau_2$ were 12 and 24 ns, respectively. The contribution of deuterium ESEEM to the PELDOR/DEER time trace was suppressed by addition of 8 observed time traces with variable $\tau_1$ time ($\Delta$ = 16 ns). The PELDOR/DEER background was fitted by a monoexponential decay. The distance distributions were calculated and validated by means of DeerAnalysis 2018[52].

**smFRET spectroscopy.** Solution based smFRET experiments were performed on a homebuilt confocal ALEX microscope as described in[15]. All sample solutions were measured with 100 μl drop on a coverslip with concentration of around 50 pM in buffer 1. The fluorescent donor molecules are excited by a diode laser OBIS 532-100-LS (Coherent, USA) at 532 nm operated at 60 μW at the sample in alternation mode (100 μs alternation period). The fluorescent acceptor molecules are excited by a diode laser OBIS 640-100-LX (Coherent, USA) at 640 nm operated at 25 μW at the sample. The lasers are combined and coupled into a polarization maintaining single-mode fibre P3-488PM- FC-2 (Thorlabs, USA). The laser light is guided into the epi-illuminated confocal microscope (Olympus IX71, Hamburg, Germany) by dual-edge beamsplitter ZT532/640rpc (Chroma/AHF) focused by a water immersion objective (UPlanSApo 60×/1.2w, Olympus Hamburg, Germany). The emitted fluorescence is collected through the objective and spatially filtered using a pinhole with 50 μm diameter and spectrally split into donor and acceptor channel by a single-edge dichroic mirror H643 LPXR (AHF). Fluorescence emission was filtered (donor: BrightLine HC 582/75 (Semrock/AHF), acceptor: Longpass 647 LP Edge Basic (Semrock/AHF), focused on avalanche photodiodes (SPCM-AQRH-64, Excelitas). The detector outputs were recorded by a NI-Card PCI-6602 (National Instruments, USA).

Data analysis was performed using home written software package as described in[33]. Single-molecule events were identified using an All-Photon-Burst-Search algorithm with a threshold of 15, a time window of 500 μs, and a minimum total photon number of 150[126]. Photon count data were extracted, background subtracted, corrected for spectral crosstalk, and different quantum yields/detection efficiencies described in[76,115]. An exemplary correction procedure of all correction steps is illustrated in Supplementary Fig. 16.

E-histogram of double-labelled FRET species with Alexa Fluor 555 and Alexa Fluor 647 was extracted by selecting $0.25 < S < 0.75$. E-histograms of open state without ligand (apo) and closed state with saturation of the ligand (holo) were fitted with a Gaussian distribution $Ae^{-\frac{(E-\mu)^2}{2\sigma^2}}$.

The burst variance analysis (BVA)[77] was performed on the same data with a photon binning of 5 photons for selected bursts with stoichiometry $0.25 < S < 0.75$.

Distance conversion was calculated according to $R = R_0 \sqrt[6]{(1-E)/E}$, where $R_0$ is the Förster radius constant. The Förster radius $R_0$ is given by[13]

$$R_0{}^6 = \frac{9\,ln(10)}{128\pi^5\,N_A}\frac{\kappa^2}{n^4}Q_D\frac{\int_0^\infty F_D(\lambda)\varepsilon_A(\lambda)\lambda^4 d\lambda}{\int_0^\infty F_D(\lambda)d\lambda}, \qquad (1)$$

where $N_A$ is the Avogadro constant, $\kappa^2$ the dipole orientation factor, and $n$ the averaged refractive index of the medium[120]. The other parameters for the donor quantum yield $Q_D$, the donor emission spectrum $F_D$, and the acceptor absorbance spectrum $\varepsilon_A$ were derived from absorption and emission spectra of singly labelled donor and acceptor mutants[72].

The overlap integral is retrieved from the emission spectrum $F_D(\lambda)$ of the donor only sample and a normalized absorption spectrum $\bar{\varepsilon}_A$ scaled to the literature extinction coefficient $\varepsilon_A(\lambda) = \varepsilon_{A_{max}}\bar{\varepsilon}_A$ according to

$$J = \frac{\int_0^\infty F_D(\lambda)\varepsilon_A(\lambda)\lambda^4 d\lambda}{\int_0^\infty F_D(\lambda)d\lambda}, \qquad (2)$$

as illustrated in Supplementary Fig. 18. The Förster radii were calculated according to Eq. (1) to be $R_0 = 51A$ for Alexa Fluor 555 – Alexa Fluor 647 and $R_0 = 52A$ for TMR – Cy5 (see Supplementary Table 5). All distance measurements were repeated three times. The individual results are listed in Supplementary Table 6. The quantum yield for TMR was experimentally determined (Supplementary Fig. 22).

**In silico distance simulations.** For both methods, we used available programmes and combined the in silico distance simulations with the experimental distances in the result parts. For PELDOR/DEER simulations, we used mtsslWizard (www.mtsslsuite.isb.ukbonn.de) where an ensemble of rotamers is calculated for each labelling position by rotation of the bonds from the spin label (see "Results")[71]. After this, the average distance and the distance distribution between two of these ensembles were determined.

For smFRET we used the FRET-restrained positioning and screening (FPS) method established by the Seidel lab[67]. This method allows the determination of a FRET-efficiency-averaged model distance between the two dyes using the crystal structure information. For distance simulations we employed a simple dye model, in which three parameters were used to determine the accessible volume the dye can sample: (i) linker-length (linker), linker-width (W), and the fluorophore volume, which can be derived from an ellipsoid using R1, R2, and R3. With this information, the average distance between two of these spheres was calculated. The dye parameter for the different fluorophores are shown in Supplementary Table 7. An average distance was calculated with the FPS software by exchanging donor and acceptor positions and vary the linker length (±1 Å) as well as linker width and radii (±0.5 Å).

**Fluorophore lifetime and time-resolved anisotropy measurements.** Lifetime and anisotropy decay measurements were performed as described in ref.[127] (ch. 11) on a homebuilt setup[72]: 400 μl of sample is measured in a 1.5 × 10 mm cuvette at a concentration of around 100 nM. The sample is excited by a pulsed laser

(LDH-P-FA-530B for green fluorophores/LDH-D-C-640 for red fluorophores with PDL 828 "Sepia II" controller, Picoquant). Excitation polarization is set with a lambda-half-waveplate (ACWP-450-650-10-2-R12 AR/AR, Laser Components) and a linear polarizer (glass polarizer #54-926, Edmund Optics). Emission light is polarization filtered (wire grid polarizer #34-315, Edmund Optics). The emission light is collected with a lens (AC254-100-A, Thorlabs) and scattering light is blocked with filters (green: 532 LP Edge Basic & 596/83 BrightLine HC, AHF; red: 635 LP Edge Basic & 685/80 ET Bandpass, AHF). The signal is recorded with an avalanche-photodiode (SPCM-AQRH-34, Excelitas) and a TCSPC module (HydraHarp400, Picoquant). Polarization optics is mounted in homebuilt, 3D-printed rotation mounts and APD is protected from scattered light with a 3D-printed shutter unit.

The excitation power was 10 μW and the concentration was finetuned to have ~50 kHz count rate under magic angle conditions. All anisotropy and lifetime measurements were recorded for 5 min in the order vertical (VV1), horizontal (VH1), magic angle (MA), horizontal (VH2), and vertical (VV2) under vertical excitation. Anisotropy was calculated based on the two vertical and horizontal measurements by taking the mean values to compensate for small drifts in laser power or slow changes in fluorophore concentration due to sticking. With $VV(t) = \frac{1}{2}(VV1(t) + VV2(t))$ and $VH(t) = \frac{1}{2}(VH1(t) + VH2(t))$, we obtain the anisotropy decay as $r(t) = \frac{VV(t) - G\,VH(t)}{VV(t) + 2G\,VH(t)}$.

**Statistics and reproducibility.** Error bars for the smFRET experiments represent the mean plus/minus the standard deviation of n=3 independent experiments. In accordance with the recent expert guidelines for large PELDOR/DEER datasets of double mutants[114] multiple double mutants of the same protein were measured and compared with structural models and the smFRET experiments to see if the ligand-induced conformational change was picked up by the set of measurements. As recommended in ref.[114], we performed multiple control measurements with different concentrations of e.g., cryoprotectants or ligand if the effect of these substances on particular samples was unexpected and merited further investigation. The control experiments are given in the supplementary information and the Source Data File. Following the guidelines in ref.[114], the uncertainty of distance distributions were determined by two different methods, Tikhonov regularization[51,52] and DeerNet[53].

No statistical method was used to predetermine sample size. No data were excluded from the analyses. The experiments were not randomized. The investigators were not blinded to allocation during experiments and outcome assessment.

**Reporting summary.** Further information on research design is available in the Nature Research Reporting Summary linked to this article.

## Data availability

Source data are provided with this paper. The EPR data generated in this study are provided in the Source Data file. The smFRET data generated in this study are available in the Zenodo database under accession code https://zenodo.org/record/6683587. The coordinate files used in this study are available in the PDB database under the following accession codes: HiSiaP open: PDB-ID: 2CEY, HiSiaP closed: PDB-ID: 3B50, MalE open: PDB-ID: 1OMP, MalE closed: PDB-ID: 1ANF, SBD2 open: PDB-ID: 4KR5, SBD2 closed: PDB-ID: 4KQP, YopO: PDB-ID: 4CI6, MMP-12: PDB-ID 5L79. Source data are provided with this paper.

## Code availability

The custom version of mtsslWizard that was used to study the impact of different linker length can be found here: https://github.com/gha2012/mtsslWizard_commandLine.

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

## Acknowledgements

This project was financed by the German Research Foundation (Deutsche Forschungsgemeinschaft, DFG) in projects no. HA 6805/4-1 and HA 6805/5-1 (to G.H.), GRK2062 project C03 (to T.C.), SFB 863 – Project number A13 – 111166240 (to T.C.), the Centre for integrated protein science Munich CiPSM (start-up funding to T.C.), an ERC starting grant ERC-StG 638536 SM-IMPORT (to T.C.), and the Centre of Nanoscience Munich (project funding to T.C.). MFP acknowledges a PhD fellowship from the Konrad Adenauer Stiftung. C.G. acknowledges a PhD fellowship from the Studienstiftung des deutschen Volkes. M.F.P., J.G., and G.H. thank Prof. Olav Schiemann, University of Bonn, for support and access to EPR spectrometers, and Dr. D. Abdullin for very valuable discussions and helpful comments on the manuscript. T.C. thanks Nicola Gericke for help with purification of substrate binding domain 2. G.H. and M.F.P. thank Dr. Frank Eggert and Prof. Stefanie Kath-Schorr for help with the MS analysis. We thank E. Lerner for the gift of HP3 and D. Griffith for proofreading and commenting on the manuscript.

## Author contributions

G.H. and T.C. conceived and supervised this study. M.F.P. and J.G. performed PELDOR/DEER experiments. C.G., G.G.M., and A.N. performed single-molecule FRET and optical spectroscopy experiments. M.F.P. and R.M. provided protein samples. M.F.P., C.G., T.C., and G.H. designed experiments and analyzed the data. M.F.P., C.G., T.C., and G.H. wrote the manuscript. All authors contributed to the discussion of the results and the final version of the manuscript.

## Funding

## Competing interests

The authors declare no competing interests.
