## [Peer Review File · Nature Communications]

Reviewers' Comments:

Reviewer #1:

Remarks to the Author:

In this manuscript a comparison of distance determination by EPR and FRET are performed on three proteins (with several mutants). Distances with and without substrate bound are compared with X-ray structure predictions. The experimental data are of very good quality and are well documented. The analysis of the FRET as well as the DEER data are performed well and the distances obtained are discussed with respect to linker length and interactions of the label with the protein. Such a comparison of two different spectroscopic methods for obtaining distance restraints is in principle interesting. Nevertheless the comparison only was qualitative - already because the linker lengths and the labels used were different for both methods. Also the comparison with the X-ray predictions is only qualitative due to the coarse grain nature of the distances obtained by both methods. Finally, the findings that interactions of the label with the protein and the flexibility of the linker limits the accuracy of both methods is not really new. Also there is no discussion about the different possibilities of both methods: to detect slow kinetics with SM-FRET on the one hand and to explore the conformational space with EPR on the other hand. Therefore I would recommend publication of this manuscript in a more specialized journal.

Reviewer #2:

Remarks to the Author:

This study addresses a very important topic. A quantitative comparison is highly appreciated. However, the study should be more comprehensive in all aspects: description of the field, design and analysis of the experiments and simulation, documentation of measurements to provide more quantitative insight and overall impact of this work.

Major comments

Abstract should be rewritten:

Line 8-10: misleading statement: "The application portfolio of the two methods is overlapping: each allows determination of distances, to monitor distance changes and to visualize conformational heterogeneity and -dynamics". I agree that EPR can monitor changes/differences for a molecule with the probe pair under distinct conditions by varying conditions, but how can measure EPR real-time dynamics in a frozen solution?

Summary corresponds to the standard statement on FRET and EPR, which is too simple as shown in this study: EPR has problems in two of three systems. The deviations between predicted and measured inter-probe distances are not assessed.

Main text:

There are several interesting aspects for the design of a comparative DEER-FRET study: (1) comparison of distances for static states (no fast exchange of the sub-millisecond time scale), (2) Influence of dynamic exchange in the sub-millisecond time range on the recovered populations and distances. The second aspect is essentially missing (only one copied data set in Fig. S5 for a dynamic DNA-hairpin). For dynamic biomolecules will be the impact of freezing more prominent. Which outcome is observed - EPR is well suited to measure distributions (see Prisner work) - thus, this case would be very interesting even for a simply biomolecule like the DNA-hairpin; (3) influence of labels (only very limited variation in this work; only FRET for limited samples and not EPR)

Literature on Integrative structural biology is not fully covered: Both seminal papers on integrative structural biology in the Journal "Structure" are missing. PDB-dev is not mentioned.

Line 25. pls add another important example: crosslinking mass spectrometry (and References)
Literature on FRET-EPR studies not fully covered. Famous groups (e.g. Jeschke, Steinhoff/Klare, Hubbell, Hummer) performed combined EPR-FRET studies and checked for consistency of their results (for the same and sometimes distinct positions)

The promise to conduct a comprehensive cross-validation is only partly fulfilled. The assessment for quantitative comparison between EPR and FRET is missing to detect potential systematic effects. Maybe a joint scatter plot of FRET distance deviation (Experimental- predicted distance) vs EPR deviation could help, to identify systematic trends, so that we could learn also something

about the studied proteins

Fig. 1F: Description of DEER experiments: red dash line is not described although it is very critical for the analysis. The determination of the slope of this line is crucial in the data analysis

Table 1: (2) homoFRET is not mentioned (e.g. work of Lennart Johansson): labels 1 or 2. Fluorescence anisotropy can be measured also on the single-molecule level.

(2) Influence of orientation effects need to be discussed (more details below)

(3) Distance range FRET: longer at least to 90- 100 Å for various regular FRET-pairs (see Atto-tec website: https://www.atto-tec.com/fileadmin/user_upload/Katalog_Flyer_Support/R_0_-_Tabelle_2018_web.pdf) and single acceptor (for used labels). For a FRET-pair with $R_0=70\text{Å}$ and $RDA=100\text{Å}$ the FRET efficiency: $E=0,105$, that can be easily measured by the Alex techniques used in this study.

Fig,2: I like the comparison. However, only on type of EPR labels is employed in this study

Fig. 7: it is not clear whether data are simulated or experimentally measured. The authors should rephrase by avoiding the word "experiment". Instead one could use the word "simulation" and simulated data....

Simulation is set up too short: there are several systematic effects that influence the experimentally recovered distances: (1) linker lengths (as simulated in this work), (2) Orientation factor at least for FRET: Linker (3) linker lengths and local concentrations that result in distinct strengths of local interactions (numerous EPR papers e.g. from the groups of Hubbell, Schiemann, Klare et. al (PLoS One)..., and FRET (Sindbert JACS 2010)), i.e. the residence time per site varies so that the integration/ averaging with the chosen time window is distinct so that the systematic deviation due to trapping is averaged out differently. To conclude, the discussion should become more comprehensive and more balanced. Finally, it is important to mention that appropriate weighting of interaction sites and rotamers is very difficult (see EPR distributions most slightly shifted between simulation) so that it could be more advantageous to have a system with lower energy barriers where averaging is faster and more easily to predict.

Moreover, the authors should add a section, where they could mention that there are solutions to this problem by inspecting the mobility of the probes. In principle this was (partly) done for both techniques (Fig. S1 (EPR) and Fig. 4 (FRET)). However, it was not done for all samples and the results were not compiled and comprehensively discussed and put in the context of the results of the distance measurements.

Anisotropy thresholds (Hellenkamp Nat. Methods. 2017) and K2 uncertainty estimates (Dale et al, van der Meer, Lakowicz, Sindbert et al.) used in FRET studies were not discussed. Moreover, orientation effects exist also in EPR - the existence and the approaches to solve this issue are not mentioned and discussed.

Moreover, other environmentally sensitive parameters like the fluorescence are not mentioned.

This would be especially relevant, because the authors used donor dyes where this is documented in many publications: Alexa-555 is equivalent to Cy3 (PIFE), TMR (in Fig. 2) (see early sm-fluorescence studies).

Quality and documentation of the experiments:

(1) Quality of the measurements: The quality of the DEER measurements have the order: HiSiaP (excellent) > SBD2 (good, partly noisy) > MalE (limited and noisy; should be improved)

.The HiSiaP measurements have high quality (long evolution time windows so that intermolecular background can be safely detected) and mostly low noise. Overall, SmFRET data are less convincing: noisy data frequently low counting statistics. SI Fig. 4, 8, 12: event number not visible, all frequencies normalized. At least in the caption the total event number used for normalization must be given.

(2) Uncertainties are missing: Fig. 3C, 4E, 5C, 6C: for FRET measurements error bars are missing

(3) FRET analysis parameters are missing: (1) FRET calibration setup parameters are missing (α , β , local-or global γ , δ ; fluorescence quantum yields $F_i(D)$ and $F_i(A)$) pls add to SI: plots for the proteins and tables with values, most importantly: no Förster radii/ distances R_0 were given for the FRET-pairs; how was R_0 for the distinct fluorescence quantum yields $F_i(D)$.

(4) the pitfalls and tricks of DEER data analysis are also not introduced, documented and discussed at all. Except discussing time window and noise, results are more or less presented as black box.

For this interdisciplinary work a comprehensive description would be very valuable

(5) Fig. 4: Fit results and quality measure for the decay analyses are completely missing (not a

reference to SI is given)

(6) Width of FRET efficiency histograms is given - but there is no discussion and comparison between molecules and dyes. What is the expected shot noise width.

Material & Methods:

Line 12: Sentence should be omitted: "For smFRET studies, optimized double cysteine mutants were created with a yet unpublished software-tool for optimal fluorophore labelling (Gebhardt & Cordes, unpublished) should be omitted. If the software is not published the subsequent statements are sufficient.

Minor comments

Abstract, line 2: Hyphen missing: single "-" molecule

Main text, line 29: allow "one" to study conformational dynamics

Page 15: cryoprotectant (Figure 5C, red traces) but in the figure are magenta traces

Reviewer #3:

Remarks to the Author:

Substrate binding and catalysis are usually connected with conformational changes of the protein. PELDOR and smFRET are methods which have been used to measure distance and changes between different sites of the protein. Both methods require attachment of labels to these sites and usually cysteines are introduced into the protein since the labels can be attached to these amino acids under physiological conditions. For PELDOR investigations spin labels are attached to these sites and the interaction between the magnetic dipoles is measured at 50 K in ensemble measurements. For FRET two different fluorescence labels are attached to these sites and the interaction between the electric dipole moments is measured at room temperature and single molecule smFRET measures these interactions within one single protein molecule. Both methods have been used frequently, however only few direct comparisons of both methods have been reported.

In this work the authors compare distance measurements in proteins carried out with PELDOR and smFRET. Three different substrate binding proteins (HiSiaP, MalE, SBD2) with known crystallographic structures of the open and closed conformations are used. Appropriate positions for double labeling of these proteins are selected from difference distance maps and the accessible volumes of the spin labels (mtssl Wizard) and FRET label (FPS) were calculated. The results of PELDOR and smFRET measurements are presented for all double mutants with and without substrates together with the corresponding simulations, all raw data are presented in the supporting informations.

The experimental distances on the same double mutant obtained with PELDOR and smFRET measurements differed by 5 Å with a spread of ± 10 Å. Considering the differences between both methods (PELDOR measures magnetic dipole interactions in a molecule ensemble at 50 K, smFRET measures electric dipole interactions in a single molecule at room temperature) this is a very good agreement.

Two sources of possible errors are identified. The differences between PELDOR and smFRET data result partly from the structures of the labels and the different linker lengths. Comparison of the experimental data with the results from simulations obtained with structural modeling programs indicate that interaction of the label with the protein can lead to wrong distance measurements. This is especially a problem for smFRET measurements due to the longer linkers and the interactions of the fluorophores with the protein. This problem was observed with the labeled HiSiaP has been investigated in detail by measurements of the time resolved fluorescence anisotropy and lifetime of the different fluorophores. A problem of PELDOR is the use of cryoprotectants influences the results of distance measurements (shown with MalE).

The experimental methods and data analysis routines for PELDOR and smFRET are state of the art. The statistical analysis is appropriate. I think it will become a standard reference in this field.

We would like to thank all referees for their time and effort to review our work. We think that the manuscript has been greatly improved both by the addition of new data as well as by the valid suggestions of the referee for text / content changes. For convenience, we provide a point-by-point response to the referee comments and queries and a document where all changes to the manuscript done during the revision are highlighted by “track changes” (changes_marked.docx).

REVIEWER COMMENTS

Reviewer #1 (Remarks to the Author):

In this manuscript a comparison of distance determination by EPR and FRET are performed on three proteins (with several mutants). Distances with and without substrate bound are compared with X-ray structure predictions. The experimental data are of very good quality and are well documented. The analysis of the FRET as well as the DEER data are performed well and the distances obtained are discussed with respect to linker length and interactions of the label with the protein.

We thank the referee for the positive evaluation of our work.

Such a comparison of two different spectroscopic methods for obtaining distance restraints is in principle interesting. Nevertheless, the comparison only was qualitative - already because the linker lengths and the labels used were different for both methods.

The referee is correct that in fluorescence and EPR techniques the nature and size of the label (and their linkers) are inherently different. Yet, in our manuscript we provide a quantitative comparison for inter-probe distances of each method with the respective simulation. We feel that this allows a direct comparison of the measurement accuracy of both methods (which we present in Figure 8). For the selected residues pairs in our proteins there is no straightforward way to correctly convert inter-probe distances to inter-residue distances, which we mention in the discussion of our manuscript. We have carefully checked the manuscript that the “comparison” aspect is communicated correctly in the revised version.

Also the comparison with the X-ray predictions is only qualitative due to the coarse grain nature of the distances obtained by both methods. Finally, the findings that interactions of the label with the protein and the flexibility of the linker limits the accuracy of both methods are indeed not really new.

While both smFRET and EPR have limitations in measurement accuracy, a comparison with structural models is one way of obtaining a ground truth for the mean inter-probe distance (e.g., based on accessible volume calculations for smFRET or rotamer libraries for EPR). In our view (and also based on community opinions e.g., in <https://elifesciences.org/articles/60416>) the comparison of an experimentally derived value to a structural model allows us to become quantitative. We agree with the referee that the finding that protein-label interactions reduce accuracy is not novel and we carefully checked the manuscript to remove statements that might suggest we are the first to describe such effects. In our view, the novel and very important contribution of our manuscript is the

comprehensive quantitative cross-validation and direct comparison of smFRET and EPR/DEER distance accuracies on the same protein systems.

Also there is no discussion about the different possibilities of both methods: to detect slow kinetics with SM-FRET on the one hand and to explore the conformational space with EPR on the other hand.

We added new data, where we compare the different possibilities of PELDOR/DEER and smFRET (in our implementation with μ sALEX with sub-millisecond temporal resolution) to study dynamic systems such as the YopO proteins' apo state. We are grateful for this comment, which was also raised by referee #2, since it provides a nice distinct angle on the interpretation of distances in the light of static and dynamic structures. We also note that we used the "standard procedure" in both EPR and smFRET for distance determination with the most widely-used probes, well-established instrumentation and data analysis methods.

Therefore, I would recommend publication of this manuscript in a more specialized journal. While we agree with most suggestions and queries of the referee to improve the quality of the paper and its scholarly presentation, the general feedback from the two other reviewers and also colleagues we discussed our results with, makes us hopeful that the improved version of the paper has a large scope in the field of integrative structural biology, including smFRET and EPR spectroscopy, and is suitable for publication in Nature Communications.

Reviewer #2 (Remarks to the Author):

This study addresses a very important topic. A quantitative comparison is highly appreciated.

We thank the referee for the positive evaluation of our work.

However, the study should be more comprehensive in all aspects: description of the field, design and analysis of the experiments and simulation, documentation of measurements to provide more quantitative insight and overall impact of this work.

We greatly appreciate this critical comment and have improved the description of field and methodologies and included missing seminal papers. We also improved our simulation, which is now based on an atomistic- instead of a geometric model. We also clearly state the limits of the simulation and we improved our discussion. We also included new experiments including a dynamic system (see below), repeated smFRET experiments with insufficient data quality and we improved the documentation of the measurements.

Major comments

Abstract should be rewritten:

Line 8-10: misleading statement: "The application portfolio of the two methods is overlapping: each allows determination of distances, to monitor distance changes and to visualize conformational heterogeneity and -dynamics". I agree that EPR can monitor changes/differences for a molecule with the probe pair under distinct conditions by varying conditions, but how can measure EPR real-time dynamics in a frozen solution?

The referee is indeed correct and this statement was removed. Clearly, detection and characterization of real-time dynamics is a unique feature of the smFRET methodology.

Summary corresponds to the standard statement on FRET and EPR, which is too simple as shown in this study: EPR has problems in two of three systems. The deviations between predicted and measured inter-probe distances are not assessed.

We have reworded the conclusion and we hope the referee agrees that it summarizes our findings more clearly now:

p. 28, bottom: "Both PELDOR/DEER and smFRET are valuable tools for the emerging toolkit of integrative structural biology. Overall, we found a reasonable agreement of the determined distances with an average ± 5 Å spread between the two methods. However, our experiments also revealed discrepancies that might have led to wrong interpretations if only one of the methods had been used. We could show that these differences were partly due to the distinct labels and label-protein interactions. Thus, reliable methods to predict or prevent label-protein interactions are urgently needed. Also, the much longer linkers used for smFRET can be problematic, while for PELDOR/DEER, the use of cryogenic temperatures and cryoprotectants was shown to influence conformational changes in an unwanted fashion. A positive outcome of our study is the observation that a combination of PELDOR/DEER and smFRET provides highly complementary and synergistic insights into the conformational states of macromolecules. Hence, the development of spectrometers and

microscopes, as well as standardized data processing approaches^{19,124}, which can also be used by non-experts, would be very beneficial to structural biology¹²⁵.”

Main text:

There are several interesting aspects for the design of a comparative DEER-FRET study: (1) comparison of distances for static states (no fast exchange of the sub-millisecond time scale),

We agree with the referee on this comment and changed the manuscript accordingly (see below).

(2) Influence of dynamic exchange in the sub-millisecond time range on the recovered populations and distances. The second aspect is essentially missing (only one copied data set in Fig. S5 for a dynamic DNA-hairpin). For dynamic biomolecules will be the impact of freezing more prominent. Which outcome is observed - EPR is well suited to measure distributions (see Prisner work) - thus, this case would be very interesting even for a simply biomolecule like the DNA-hairpin;

Thank you for this comment and the suggestion to include a dynamic (protein) system. We changed the manuscript in two different ways: (i) we have now cited the literature where attempts at uncovering conformational dynamics by EPR were made. (ii) In addition, we have added a set of new experiments on the YopO protein, where broad distance distributions were observed in the protein apo state with PELDOR/DEER in our previous publication in Structure (Peter et al. 2018). We have now also studied this system with smFRET in direct comparison to PELDOR/DEER. This suggestion led to an excellent example to illustrate the different points of views of the two techniques: bulk vs single molecule and the resulting possibility to study dynamics with smFRET. Hence, we added a new section to the manuscript as shown in new Figure 7. We note that we have not conducted experiments based on pulsed laser excitation (MFD/PIE) since this would have not been possible with the most common dye pair Alexa555/647 due to the short donor-lifetime.

(3) influence of labels (only very limited variation in this work; only FRET for limited samples and not EPR)

We agree with the referee that our emphasis was not on the variation of the labels for one protein system, but our intention was to use established “probes” for both methods and to start a comparison on different protein systems. We consider this a useful strategy since other researchers would try first exactly this, when a new protein system is investigated. Furthermore, we do show in Figure 4 (and associated SI Figures) how fluorescent labels impact the obtained smFRET results. For PELDOR/DEER we decided to focus solely on the MTSSL label, which is clearly by far the most used, commercially available and characterized label. Overall, we think that a systematic comparison of different EPR or FRET labels is out of the scope of our manuscript. This is also phrased in the revised text as follows:

p. 8 bottom: “Our aim for the comparison was to choose commonly used labels and well-established experimental conditions for either method. The PELDOR experiments were thus performed at 50 K with cryo-protected samples using a commercial pulsed Q-band

spectrometer. The samples were labelled with MTSSL (S-(1-oxyl-2,2,5,5-tetramethyl-2,5-dihydro-1H-pyrrol-3-yl)methyl methanesulfonothioate)⁷⁰. The distance distributions were determined using DeerAnalysis⁵⁰ and the distance predictions were calculated with mtsslWizard⁷⁴. The smFRET experiments of diffusing protein molecules were conducted in buffer at room temperature using standard procedures suitable for microsecond alternating laser excitation (ALEX) as described before⁶⁸. The experiments were performed with Alexa Fluor 555 (donor) and Alexa Fluor 647 (acceptor) using a homebuilt confocal microscopy setup with 2-colour excitation/detection⁶⁸.”

Literature on Integrative structural biology is not fully covered: Both seminal papers on integrative structural biology in the Journal "Structure" are missing. PDB-dev is not mentioned.

We thank the referee to note this important point; we added the missing citations and also the PDB-Dev is now mentioned in the paper:

p.3, bottom: “Solutions for the nontrivial task of archiving the resulting hybrid models, which are often based on multiple datasets from very different experimental techniques, are developed by the PDB-Dev initiative⁹.

.”

Line 25. pls add another important example: crosslinking mass spectrometry (and References)

Done.

Literature on FRET-EPR studies not fully covered. Famous groups (e.g. Jeschke, Steinhoff/Klare, Hubbell, Hummer) performed combined EPR-FRET studies and checked for consistency of their results (for the same and sometimes distinct positions)

We added additional citations that cover – to the best of our knowledge – the relevant papers.

The promise to conduct a comprehensive cross-validation is only partly fulfilled. The assessment for quantitative comparison between EPR and FRET is missing to detect potential systematic effects. Maybe a joint scatter plot of FRET distance deviation (Experimental- predicted distance) vs EPR deviation could help, to identify systematic trends, so that we could learn also something about the studied proteins

We agree with the referee that this is a useful analysis direction. Indeed, we have done this type of analysis in former Figure 7C (and the SI), which is now found in Figure 8. To emphasize this important aspect, we improved the related discussion section.

Fig. 1F: Description of DEER experiments: red dash line is not described although it is very critical for the analysis. The determination of the slope of this line is crucial in the data analysis

We improved the description of DEER experimental work flow and data analysis. This is now explained in the discussion section.

Table 1: (2) homoFRET is not mentioned (e.g. work of Lennart Johansson): labels 1 or 2. Fluorescence anisotropy can be measured also on the single-molecule level.

We agree that we should have cited and mentioned this possibility and extended the description (and appropriate citation) in Table 1. Further discussion see also point (3) below.

(2) Influence of orientation effects need to be discussed (more details below)

We added a text section on this aspect in the discussion section, where we also outline the possibility to use accessible contact volumes for the fluorophores instead of accessible volumes to account for protein fluorophore interactions. Orientation selectivity is also mentioned for PELDOR/DEER in the discussion.

(3) Distance range FRET: longer at least to 90-100 Å for various regular FRET-pairs (see Atto-tec website: https://www.atto-tec.com/fileadmin/user_upload/Katalog_Flyer_Support/R_0_-Tabelle_2018_web.pdf) and

single acceptor (for used labels). For a FRET-pair with $R_0=70\text{Å}$ and $R_{DA}=100\text{Å}$ the FRET efficiency: $E=0,105$, that can be easily measured by the Alex techniques used in this study.

We agree with the referee that it is possible to do these types of measurements with two (spectrally) identical fluorophores. We also emphasize, however, that they are less common in the community and we decided to focus the table on the "gold-standard", i.e., single-pair / 2-colour FRET. We also looked through the table on the ATTO-page and found various combinations with Förster radii around 7 nm as stated by the referee. Yet, these refer to dye combinations that are not well characterized in relation to single-molecule detection (e.g., Rho 11, Rho 12, Rho 101 as donor dyes), red- nearIR combinations such as donor ATTO643/ATTO647N with ATTO700 (for which most groups do not have suitable excitation/detection schemes) or homoFRET approaches (which require time-resolved analysis of fluorescence and cannot use μsALEX).

Fig,2: I like the comparison. However, only on type of EPR labels is employed in this study

As stated above, we decided to use the standard label (MTSSL) for our comparative study, but decided to provide an overview of EPR-labels that are under active development.

Besides commercial unavailability most other spin labels are still not yet fully optimized for chemical stability, optimal behavior after attachment to a protein and lack standardized measurement/analysis of the corresponding EPR experiment.

Fig. 7: it is not clear whether data are simulated or experimentally measured. The authors should rephrase by avoiding the word "experiment". Instead one could use the word "simulation" and simulated data....

Done.

Simulation is set up too short: there are several systematic effects that influence the experimentally recovered distances:

(1) linker lengths (as simulated in this work),

(2) Orientation factor at least for FRET: Linker

(3) linker lengths and local concentrations that result in distinct strengths of local interactions (numerous EPR papers e.g. from the groups of Hubbell, Schiemann, Klare et al (PLoS One)..., and FRET (Sindbert JACS 2010)), i.e. the residence time per site varies so that the integration/ averaging with the chosen time window is distinct so that the systematic deviation due to trapping is averaged out differently.

To conclude, the discussion should become more comprehensive and more balanced.

Finally, it is important to mention that appropriate weighting of interaction sites and rotamers is very difficult (see EPR distributions most slightly shifted between simulation) so that it could be more advantageous to have a system with lower energy barriers where averaging is faster and more easily to predict.

In the revised version, our simulation is now based on actual atomic models of the labelled HiSiaP protein, the MTSSL label and the Alexa Fluor 647 label instead of simple geometric models as before. While the overall message stays the same, the outcome is now easier to interpret and the simulation now contains information about the relative orientation of ensembles. We have included a note that for smFRET, the geometric models cannot be directly compared to the distance that is extracted from experimental data and clarified that we are only considering the geometric models. We also extended and streamlined the discussion section into various parts that we hope convey the message clearer.

Moreover, the authors should add a section, where they could mention that there are solutions to this problem by inspecting the mobility of the probes. In principle this was (partly) done for both techniques (Fig. S1 (EPR) and Fig. 4 (FRET)). However, it was not done for all samples and the results were not compiled and comprehensively discussed and put in the context of the results of the distance measurements.

We have added a section to the discussion (page 23) that provides details to this point. We have decided not to add new data, since a general addition of "mobility" data of all label positions is out of the scope of our study.

p. 23 bottom: "Luckily, there are experimental approaches to detect strong protein label interactions in smFRET (Figure 4) as well as for PELDOR/DEER. For the latter, the shape of room temperature cw spectra and abnormally shaped Pake patterns can be used to detect strongly immobilized spin labels^{96,97}. Depending on the label, its degree of immobilization and especially at high magnetic fields, it can be necessary to consider orientation selection effects by collecting multiple PELDOR/DEER time traces at different frequency offsets⁹⁷⁻⁹⁹."

Anisotropy thresholds (Hellenkamp Nat. Methods. 2017) and K2 uncertainty estimates (Dale et al, van der Meer, Lakowicz, Sindbert et al.) used in FRET studies were not discussed.

We have added a dedicated section to "smFRET-specific problems" (page 26) to discuss these problems and thank the referee for pointing this out.

Moreover, orientation effects exist also in EPR - the existence and the approaches to solve this issue are not mentioned and discussed.

This is now mentioned in the manuscript.

p. 23 bottom: “Luckily, there are experimental approaches to detect strong protein label interactions in smFRET (Figure 4) as well as for PELDOR/DEER. For the latter, the shape of room temperature cw spectra and abnormally shaped Pake patterns can be used to detect strongly immobilized spin labels^{96,97}. Depending on the label, its degree of immobilization and especially at high magnetic fields, it can be necessary to consider orientation selection effects by collecting multiple PELDOR/DEER time traces at different frequency offsets⁹⁷⁻⁹⁹.”

Moreover, other environmentally sensitive parameters like the fluorescence are not mentioned. This would be especially relevant, because the authors used donor dyes where this is documented in many publications: Alexa-555 is equivalent to Cy3 (PIFE), TMR (in Fig. 2) (see early sm-fluorescence studies).

We fully agree with the referee that Alexa 555 is equivalent to Cy3 and we thus expect that PIFE effects can alter the Förster radius but also gamma in our correction procedure and with that there might be impacts on the distance accuracy. We inserted a discussion point on this in the section “smFRET-specific problems”.

Quality and documentation of the experiments:

(1) Quality of the measurements: The quality of the DEER measurements have the order: HiSiaP (excellent) > SBD2 (good, partly noisy) > MalE (limited and noisy; should be improved). The HiSiaP measurements have high quality (long evolution time windows so that intermolecular background can be safely detected) and mostly low noise.

We now explain in the manuscript, i.e., in the workflow description of the methods and in the dedicated methods section, why the data quality between the examples is different and that it is not based on arbitrary decisions. We now provide 2-pulse ESEEM experiments of our samples to rationalize the quality of the DEER measurements for the different proteins. This demonstrates the tradeoff between signal-to-noise ratio of the data and length of the time window and this is now also explained in the text. In the case of MalE the 2pESEEM experiments illustrate that the t_2 relaxation times of the samples are shorter than for the other proteins and hence the signal at long dipolar evolution times was worse than for the other proteins. The referee noted the importance of being able to fit the intermolecular background and this was the reason why we went for a longer time window and at the same time inevitably sacrificed signal-to-noise ratio. The MalE data is clearly of sufficient quality for the conclusions that are drawn from them and the data quality is on par with the current literature on PELDOR/DEER on proteins. We agree that the 87/127 mutant is an exception but this is clearly mentioned in the text. We think that not excluding the non-optimal performing mutants adds to the manuscript.

p. 14, middle: “The phase memory times of the MalE samples were significantly shorter than for the HiSiaP samples (SI Figure 17). Nevertheless, it was possible to measure time traces with sufficient length to resolve the expected distances, albeit at a lower SNR compared to the HiSiaP samples (see Discussion and SI Figure 9). Three of the four MalE variants yielded

good-quality PELDOR time traces. The 87/127 variant had a relatively low modulation depth and SNR but still provided data of sufficient quality (SI Figure 9).”

p.26, bottom: “A sufficient length of the time trace is important, because otherwise, the crucial procedure of fitting and removing the intermolecular background (red dashed line Figure 1F) as well as determining the dampening of the oscillations quickly becomes arbitrary¹¹¹. Hence, a tradeoff between signal strength and length of the time-window must be made for each sample. It is usually more important to have a longer time window than high SNR. “

Overall, smFRET data are less convincing: noisy data frequently low counting statistics. SI Fig. 4, 8, 12: event number not visible, all frequencies normalized. At least in the caption the total event number used for normalization must be given. (2) Uncertainties are missing: Fig. 3C, 4E, 5C, 6C: for FRET measurements error bars are missing

We agree with the reviewer that some data in the previous version had lower counting statistics. Therefore, we repeated the measurements on HiSiaP (Figure 3 + SI Figure 4) with better statistics but similar results. We further added the event number to the figure captions for all smFRET experiments as suggested.

(3) FRET analysis parameters are missing: (1) FRET calibration setup parameters are missing (alpha, beta, local-or global gamma, delta; fluorescence quantum yields $F_i(D)$ and $F_i(A)$) pls add to SI: plots for the proteins and tables with values, most importantly: no Förster radii/distances R_0 were given for the FRET-pairs; how was R_0 for the distinct fluorescence quantum yields $F_i(D)$.

We thank the referee for noting this and inserted a new SI Table summarizing all parameters used for the correction of apparent FRET efficiencies. (Table S1). We further added a paragraph to the methods part, where we explain the calculation of the Förster radius and provide the relevant numbers (Table S2).

(4) the pitfalls and tricks of DEER data analysis are also not introduced, documented and discussed at all. Except discussing time window and noise, results are more less presented as black box. For this interdisciplinary work a comprehensive description would be very valuable

We agree with the referee and added a short description at the beginning of the results and also a dedicated section in the discussion: p.6 and p.24

(5) Fig. 4: Fit results and quality measure for the decay analyses are completely missing (not a reference to SI is given)

We note here that the presented anisotropy and lifetime decays in Figure 4 do not include any fits. Thus, fit results cannot be given. We think that a quantitative analysis of the decays does not add more insights on our results and possible interpretations.

(6) Width of FRET efficiency histograms is given - but there is no discussion and comparison between molecules and dyes. What is the expected shot noise width.

We thank the referee for this comment. It was really missing and useful information that is now used at various places in the new manuscript. The data is integrated in Table 2.

Material & Methods:

Line 12: Sentence should be omitted: "For smFRET studies, optimized double cysteine mutants were created with a yet unpublished software-tool for optimal fluorophore labelling (Gebhardt & Cordes, unpublished) should be omitted. If the software is not published the subsequent statements are sufficient.

We agree with the referee and removed the statement.

Minor comments

Abstract, line 2: Hyphen missing: single "-" molecule

Corrected.

Main text, line 29: allow "one" to study conformational dynamics

We have added the word "scientists".

Page 15: cryoprotectant (Figure 5C, red traces) but in the figure are magenta traces

Corrected.

Reviewer #3 (Remarks to the Author):

Substrate binding and catalysis are usually connected with conformational changes of the protein. PELDOR and smFRET are methods which have been used to measure distance and changes between different sites of the protein. Both methods require attachment of labels to these sites and usually cysteines are introduced into the protein since the labels can be attached to these amino acids under physiological conditions. For PELDOR investigations spin labels are attached to these sites and the interaction between the magnetic dipoles is measured at 50 K in ensemble measurements. For FRET two different fluorescence labels are attached to these sites and the interaction between the electric dipole moments is measured at room temperature and single molecule smFRET measures these interactions within one single protein molecule. Both methods have been used frequently, however only few direct comparisons of both methods have been reported.

In this work the authors compare distance measurements in proteins carried out with PELDOR and smFRET. Three different substrate binding proteins (HiSiaP, MalE, SBD2) with known crystallographic structures of the open and closed conformations are used.

Appropriate positions for double labeling of these proteins are selected from difference distance maps and the accessible volumes of the spin labels (mtssl Wizard) and FRET label (FPS) were calculated. The results of PELDOR and smFRET measurements are presented for all double mutants with and without substrates together with the corresponding simulations, all raw data are presented in the supporting informations.

The experimental distances on the same double mutant obtained with PELDOR and smFRET measurements differed by 5 Å with a spread of ± 10 Å. Considering the differences between both methods (PELDOR measures magnetic dipole interactions in a molecule ensemble at 50 K, smFRET measures electric dipole interactions in a single molecule at room temperature) this is a very good agreement.

Two sources of possible errors are identified. The differences between PELDOR and smFRET data result partly from the structures of the labels and the different linker lengths.

Comparison of the experimental data with the results from simulations obtained with structural modeling programs indicate that interaction of the label with the protein can lead to wrong distance measurements. This is especially a problem for smFRET measurements due to the longer linkers and the interactions of the fluorophores with the protein. This problem was observed with the labeled HiSiaP has been investigated in detail by measurements of the time resolved fluorescence anisotropy and lifetime of the different fluorophores. A problem of PELDOR is the use of cryo-protectants influences the results of distance measurements (shown with MalE).

The experimental methods and data analysis routines for PELDOR and smFRET are state of the art. The statistical analysis is appropriate. I think it will become a standard reference in this field.

Thank you very much for this very positive review of our manuscript!

Reviewers' Comments:

Reviewer #1:

Remarks to the Author:

The authors have very much improved and extended their manuscript based on the remarks and suggestions of two reviews of the original submission. In the present form I believe that it is a very nice quantitative comparison between both methods - with x-ray structures and should be published now in Nature Communications.

Reviewer #2:

Remarks to the Author:

The authors try to provide a quantitative study of EPR and smFRET data, but the manuscript still does not provide sufficient depth in the pitfalls of techniques with respect to data analysis and error sources (simple things that one finds in text books are explained and the decisive details are still missing (e.g. Fig.1). To refer to further literature does not help to bridge communities.

Major comments:

Page 10:(Fig. 3) "The red shade around the PELDOR/DEER data is the error margin calculated using the validation tool of DeerAnalysis". The authors should explain/mention what the error tool does: Is this precision as usual? Obviously, the main problem is the accuracy that is not discussed at this stage (a reference to the discussion in Fig. 7 would be useful). One major reason for the limited accuracy is the uncertainty which rotamers are actually populated.

smFRET: the choice of Alexa555 and TMR should not be advised for quantitative FRET-based structural studies because its fluorescence is highly sensitive on the environment (PIFE) strongly affecting R_0 , which is confirmed by the authors. However, I agree that due to its good photostability it can be very useful for kinetic studies with qualitative structural information. However, in this work, the main focus is structural biology - thus, the choice of the donor dye is not fortunate. Moreover, the authors did not analyse the fluorescence parameters of the donor and acceptor in Fig. 4 (I disagree with authors here, because they assess and compare the techniques without performing a careful analysis of the label properties). Thus, the data do not have the required quality for the comparison. Moreover, the molecular insight is limited and more descriptive, which is not sufficient for a publication in a high-impact journal such as Nat. Commun. No wonder that there can be differences in distinct directions. The authors mention small changes of the dye properties on page 13, but whether this finding was considered in the distance calculations remains unclear.

Another example. SI Table 2. The fluorescence quantum yield of TMR (0.11) seems to be wrong. The authors state that it behaved like in water (here the fluorescence quantum yield is in the range of approx. 50%). Thus, the Förster Radius should be corrected.

Fig 7D The predicted distance for FRET are hardly visible.

Fig 7E BVA principle: Panel two "static" case is misleading, because two scenarios should be considered: static and very fast (with relaxation times < approx. 200 μ s).

Fig. 8 C,D. The way the data are plotted makes a rigorous analysis impossible. (1) The experiment number should be assigned to the actual experiments, so that the deviations can be backtracked. Moreover, the authors could provide a table with all numbers. (2) data should be plotted "measured" vs "predicted" so that the absolute values are visible (by plotting differences this is invisible). In the current way of presentation, the sensitive range of the techniques is hidden (therefore it is unusual in the field).

Uncertainties (error bars) are still missing for Fig. 3C, 4E, 5C, 6C for FRET measurements, because no error propagation (especially in R_0 (see Hellenkamp et al. Nat. Methods) and precision analysis was performed. This results in qualitative discussion of the results, which does not satisfy the title of manuscript "cross-validation of distances...", without knowing the uncertainties of the experimentally determined distances.

Old Fig 7C and New Fig. 8 C+D: differ, but it is unclear what has been changed. In the new

version there are obviously less measurements. It seems that the authors have removed data instead of remeasuring them.

Overall, I must conclude that the manuscript should not be published in its present form.

Minor comments:

Page 3 (Line 28-31): The description of PDB-dev is not appropriate. Please have a look at their webpage for a more appropriate description "PDB-Dev is a prototype archiving system for structural models obtained using integrative or hybrid modeling and is funded by the NSF ABI Development Program".

We thank the referees for their time and effort that improved the quality of manuscript.

Reviewer #1 (Remarks to the Author):

The authors have very much improved and extended their manuscript based on the remarks and suggestions of two reviews of the original submission. In the present form I believe that it is a very nice quantitative comparison between both methods - with x-ray structures and should be published now in Nature Communications.

Thank you very much for this positive review of our revised version!

Reviewer #2 (Remarks to the Author):

The authors try to provide a quantitative study of EPR and smFRET data, but the manuscript still does not provide sufficient depth in the pitfalls of techniques with respect to data analysis and error sources (simple things that one finds in text books are explained and the decisive details are still missing (e. g. Fig.1). To refer to further literature does not help to bridge communities.

We thank the referee for the suggestion to discuss the pitfalls and error sources more thoroughly. The determination of error margins is now thoroughly discussed in the method specific sections of the discussion (see points below).

Major comments:

Page 10:(Fig. 3) "The red shade around the PELDOR/DEER data is the error margin calculated using the validation tool of DeerAnalysis". The authors should explain/mention what the error tool does:

We have added a reference to the extended Discussion in the caption of all relevant figures.

E. g. caption of Figure 3:

"The underlying principle of the validation tool is explained in the discussion section below."

Further, the different validation approaches are now explained in the discussion and we have added DeerNet validations to the supplemental data (SI Figure 19/20):

p25, last paragraph: "It is a strong point of PELDOR/DEER that distance distributions rather than average distances can be readily obtained. However, it is important to remember that the shape of these distributions depends on a number of parameters such as the quality and especially the length of the underlying PELDOR/DEER time trace^{45,114}. Unfortunately, its maximum length cannot be arbitrarily chosen, since the refocused echo signal quickly decreases with an increasing time window (Figure 1D). Hence, a tradeoff between signal strength and length of the time-window must be made for each sample, where it is usually more important to have a longer time window than a high SNR. The conversion of time traces into distance distributions is often solved by a two-step analysis of first fitting and removing the intermolecular background (Figure 1D, dashed red line) and then applying Tikhonov regularization to extract the distance information^{51,52}. The procedure introduces a regularization parameter α that describes a compromise between the smoothness of the distance distribution and how well it reproduces the experimental time-trace⁵¹. Because the true shape of the distribution is of course unknown, this procedure inevitably introduces a degree of uncertainty. The two-step procedure works well for high-quality data, where more than a complete oscillation period of the signal was recorded (e.g. SI Figure 2). In practice, this is not always the case and the separation of the intermolecular background becomes a source of uncertainty. The evaluation feature of the DeerAnalysis software can be used to visualize the impact of this problem on the distance distribution. The software systematically varies parameters, such as the starting time of the background fit to obtain a mean value, a

standard deviation, as well as upper- and lower limits for each point of the distance distribution¹¹⁵. In our comparisons above, the red shade around the distance distribution was produced with this feature. Recently, new data processing algorithms have been developed that for example calculate the distance distribution in a more robust one-step analysis and also consider the noise level in the raw data to estimate the uncertainty of the distance distribution^{116,117}. Yet another approach is DeerNet^{53,118}, an artificial neural network that was trained on a large database of simulated data. This latter method is independent of user-adjustable parameters. As a comparison, we processed the key datasets measured in this study with DeerNet and reassuringly found very similar results SI Figure 19, 20.

In a recent study, aliquots of the same PELDOR/DEER samples were analyzed by seven EPR laboratories¹¹⁹. While the resulting distance distributions were overall quite similar, the variation between the individual labs was not fully covered by the error margins calculated with the different processing algorithms. It was concluded that this was caused by the uncertainty of background separation due to the different lengths of the time traces that were recorded by the different labs and an overlap of the excitation bands of observer- and pump pulses which is not yet accounted for by the algorithms^{116,119}.

Is this precision as usual?

The HiSiaP data is of very high quality, as mentioned in the text. The precision is of course dependent on the data quality. We have clarified this in the main text:

P9, first paragraph: “the PELDOR/DEER-time traces obtained for HiSiaP were of excellent quality with clearly visible oscillations and high signal to noise ratios (SNR, SI Figure 2). The distance distributions had a single, well-defined peak (Figure 3C, black curves) with very small uncertainties (red shades) and a clear shift towards shorter distances in the presence of substrate.”

Obviously, the main problem is the accuracy that is not discussed at this stage (a reference to the discussion in Fig. 7 would be useful). One major reason for the limited accuracy is the uncertainty which rotamers are actually populated.

We have added this point and a reference to the discussion.

P9, first paragraph: “...were in good agreement with the experimental PELDOR/DEER results, considering the known error of $\pm 2-4$ Å for such predictions. This error margin is mainly due to difficulties in correctly predicting the conformation of the spin label, as discussed below and in⁷⁶. “

smFRET: the choice of Alexa555 and TMR should not be advised for quantitative FRET-based structural studies because its fluorescence is highly sensitive on the environment (PIFE) strongly affecting R0, which is confirmed by the authors. However, I agree that due to its good photostability it can be very useful for kinetic studies with qualitative structural information. However, in this work, the main focus is structural biology - thus, the choice of the donor dye is not fortunate.

We agree with the referee (and mention this in the paper) that Alexa Fluor 555 is a dye that can show environmentally dependent fluorescence parameters. For TMR (as a structurally distinct rhodamine dye) this is not so much the case. We stress that our goal was to provide a comparison of smFRET and EPR using the most commonly used labels. The idea was to allow many users to adapt the analysis routines, e.g., for obtaining accurate FRET values and distances. And the most commonly used donor dye for smFRET is Alexa Fluor 555.

Furthermore, our study shows a very good agreement between smFRET and EPR in both a qualitative (trends of distance changes) and quantitative sense (distance accuracy in

comparison to AV calculations for smFRET or rotamer libraries for EPR). In the instances where discrepancies were observed between the techniques (or simulations), we managed to identify the reason for it. So, the main message of our work is very clear: Both methods deliver reliable and synergistic results and good accuracy. But of course, pitfalls such as unwanted protein-label interactions exist, yet, we believe that there is currently no clear way to predict when fluorescence/spin labels show this type of behaviour.

To validate the idea that Alexa Fluor 555 (as a sulfonated equivalent of Cy3) is as useful as other dyes for quantitative smFRET, we compared smFRET-derived distances of Alexa Fluor 647 when using Alexa Fluor 555 (cyanine), 532 and 546 (both rhodamine dyes). We have now integrated a new SI Figure and Table, which contains 4 new experimental data sets with both new donor fluorophores (Alexa Fluor 532 and Alexa Fluor 546) tested on MalE and SBD2; see new SI Figure 21. The data (A/B) and a plot of ΔR -values (theoretical vs. experimental interprobe distance) clearly shows that the deviations are very similar for all dye pairs. Importantly we can see that certain labelling positions show larger deviations due to sticking of the fluorophores, which also seems independent on the choice of dye.

In summary, we share the concern of the referee that Alexa Fluor 555 does undergo PIFE which is position-dependent, yet our results clearly demonstrate that the distance accuracy is more governed by the selected labelling position than the dye itself. The figure content was inserted into the manuscript as new SI Figure and into the discussion section.

SI Figure 21: smFRET data of MalE with Alexa Fluor 532 – Alexa Fluor 647 and SBD with Alexa Fluor 546 – Alexa Fluor 647. A) ES-2D-Histograms of MalE variant 29/352 in apo/holo state and variant 87/186 in apo/holo state (left to right). The numbers of considered bursts N are 1532/863/1286/733 (left to right). **B)** ES-2D-Histograms of SBD2 variant 369/392 in apo/holo state and variant 319/451 in apo/holo state (left to right). The numbers of considered bursts N are 559/919/1509/1108 (left to right). **C)** Deviation of calculated

distance and simulated distance for MalE (left) and SBD2 (right) for the four measurement conditions show only small variations (<2 Å) for different fluorophores (see Table 3).

SI Table 4: Fluorophore comparison. Overview of simulated and measured distances with varying donor fluorophores for selected mutants

	Simulated distance	Measured distance	Simulated distance	Measured distance
Mutant	Alexa Fluor 555 – Alexa Fluor 647		Alexa Fluor 532 – Alexa Fluor 647*	
MalE 29/352, apo	71.1	63.7	72.0	66.1
MalE 29/352, holo	57.7	56.2	59.1	59.7
MalE 87/186, apo	47.5	48.2	50.2	52.1
MalE 87/186, holo	54.2	54.6	57.2	58.2
Mutant	Alexa Fluor 555 – Alexa Fluor 647		Alexa Fluor 546 – Alexa Fluor 647**	
SBD2 369/451, apo	65.1	60.7	65.4	60.8
SBD2 369/451, holo	52.1	51.9	51.2	49.3
SBD2 319/392, apo	58.3	54.5	57.6	56
SBD2 319/392, holo	44.0	45.3	44.2	45.8

*Förster radius: $R_0=61$ Å

** Förster radius: $R_0=66$ Å

Moreover, the authors did not analyse the fluorescence parameters of the donor and acceptor in Fig. 4 (I disagree with authors here, because they assess and compare the techniques without performing a careful analysis of the label properties). Thus, the data do not have the required quality for the comparison. Moreover, the molecular insight is limited and more descriptive, which is not sufficient for a publication in a high-impact journal such as Nat. Commun. No wonder that there can be differences in distinct directions. The authors mention small changes of the dye properties on page 13, but whether this finding was considered in the distance calculations remains unclear.

The referee is correct that we did not compare the fluorescence parameters of all proteins and all label positions, which is a project on its own and not the subject of this paper. Instead, we used published values for QY and spectral properties of the dyes to make a distance-error estimation that considers donor-QY, overlap integral and the orientation factor of the donor and acceptor. The limitations of our error analysis are now discussed in the discussion section. We believe that the discussion section and the high quality of our smFRET results show that intensity-based smFRET investigations are sufficient to obtain accurate distances and even the use of literature values for spectroscopic parameters does not limit use of smFRET investigations for accurate distance determination. So, we believe this point even argues in our favour since the results show the robust nature of the smFRET approach and analysis.

Another example. SI Table 2. The fluorescence quantum yield of TMR (0.11) seems to be wrong. The authors state that it behaved like in water (here the fluorescence quantum yield is in the range of approx. 50%). Thus, the Förster Radius should be corrected.

To the best of our knowledge the applied Förster Radius is around 5.0 nm (+/- 0.4 nm) with one exception according to various publications (as listed below). This is further consistent with the supplier-based quantum yield of 0.1 for TMR: <https://de.lumiprobe.com/p/tamra-maleimide-5> (independent of the coupling chemistry group that is attached).

QY	R0 TMR-Cy5 [A]	Source
	53	https://www.pnas.org/content/97/10/5179#ref-16 https://www.pnas.org/content/96/3/893?ikey=dd270580739c311b0d59d25ba4260efb7da06de7&keytype=tf_ipsecsha
	53	https://www.koreascience.or.kr/article/JAKO202116739361203.pdf
	65	http://www.rowland.harvard.edu/labs/meller/graphics/FRETNonIdealTransfer.pdf
0.10	46	https://www.sciencedirect.com/science/article/pii/S0003267005021094?via%3Dihub
0.1		https://de.lumiprobe.com/p/tamra-maleimide-5
0.1-0.11		https://onlinelibrary.wiley.com/doi/full/10.1111/i.1751-1097.2005.tb00244.x

Fig 7D The predicted distance for FRET are hardly visible.
We have improved the visibility of the predicted FRET distance.

Fig 7E BVA principle: Panel two "static" case is misleading, because two scenarios should be considered: static and very fast (with relaxation times < approx. 200 μ s).
We thank the referee for noting this and added an additional description for panel E of Figure 7: "... (right); please note that exchange of conformational states on timescales much faster than 100 μ s can also give rise to an apparent static behaviour of the burst in BVA."

Fig. 8 C, D. The way the data are plotted makes a rigorous analysis impossible. (1) The experiment number should be assigned to the actual experiments, so that the deviations can be backtracked. Moreover, the authors could provide a table with all numbers. (2) data should be plotted "measured" vs "predicted" so that the absolute values are visible (by plotting differences this is invisible). In the current way of presentation, the sensitive range of the techniques is hidden (therefore it is unusual in the field).

We have modified Figure 8 In the following way:

- Figure 8C is now the requested "measured" vs "predicted" plot that shows the range of distances that have been measured.
- The old panels 8CD are now 8DE and the x-axis in these two panels now represents the number of the experiment in Table S3, which contains all plotted data. Thus, the data can now be fully backtracked.

SI Table 2: Details for the measurements plotted in Figure 8CDE.

Exp.	Sample	FRET Exp (Å)	FRET sim (Å)	PELDOR exp (Å)	PELDOR sim (Å)	PELDOR exp vs. PELDOR sim (Å)	FRET exp vs. FRET sim (Å)	FRET exp vs. PELDOR exp (Å)	FRET sim vs. PELDOR sim (Å)
1	HiSiaP 58/134 apo	63.6	65.6	59	62.5	-3.5	-2	4.6	3.1
2	HiSiaP 58/134 holo	57.9	59.8	56	55	1	-1.9	1.9	4.8
3	HiSiaP 55/175 apo	43	45.2	42.6	38.1	4.5	-2.1	0.4	7.1
4	HiSiaP 55/175 holo	38.7	35.8	27.8	24.3	3.5	2.9	10.9	11.5
5	HiSiaP 112/175 apo	55.1	59.3	58.4	56.5	1.9	-4.2	-3.3	2.8
6	HiSiaP 112/175 holo	56.9	52.1	50.4	46.5	3.9	4.8	6.5	5.6
7	HiSiaP 175/228 apo	59.6	65.7	59.5	62.3	-2.8	-6.1	0.1	3.4
8	HiSiaP 175/228 holo	60.6	57.8	50.1	52.4	-2.3	2.8	10.5	5.4
9	MalE 134/186 apo	42	37.9	25.2	28.4	-3.2	4.1	16.8	9.5
10	MalE 29/352 apo	63.7	71.1	62.3	67.8	-5.6	-7.4	1.5	3.3
11	MalE 36/352 apo	56.5	58.8	56.5	57.5	-1	-2.3	0	1.3
12	MalE 87/127 apo	44.1	41.5	29.4	31.4	-2	2.6	14.7	10.1
13	MalE 87/127 holo	50.1	48.3	36.4	39.1	-2.7	1.8	13.7	9.2
14	SBD2 319/392_holo	45.3	44	42.3	38.9	3.7	1.3	3.0	5.4
15	SBD2 369/451 holo	51.9	52.1	44	47.6	-3.6	-0.2	7.9	4.5

Uncertainties (error bars) are still missing for Fig. 3C, 4E, 5C, 6C for FRET measurements, because no error propagation (especially in R0 (see Hellenkamp et al. Nat. Methods)) and precision analysis was performed. This results in qualitative discussion of the results, which

does not satisfy the title of manuscript "cross- validation of distances...", without knowing the uncertainties of the experimentally determined distances.

We performed error estimation and distance error propagation according to Hellenkamp et al. but we did not include error values in the figures for FRET measurements. We conclude that all distances are in the range of $0.8 * R_0$ to $1.3 * R_0$ and thus background, α , and δ errors play a minor role ($\Delta R < 1 \text{ \AA}$) as suggested Hellenkamp et al. Therefore, the major contribution towards errors in smFRET-based distances are based on wrong R_0 determination or incorrect γ -values. We estimate $\Delta R_\gamma \approx 1 \text{ \AA}$ and $\Delta R_{R_0} \approx 3 - 4.5 \text{ \AA}$ (depending on the distance) based on a relative error in γ and R_0 of 10% and 7%, respectively. For HiSiaP-mutants including the problematic residue 175, larger values are expected. We added a new section describing the different error contributions. Based on these considerations we estimate the errors as follows

delta R for different distances	
R [Å]	deltaR [Å]
45	2.8
50	3.2
55	3.5
60	3.9
65	4.4
70	5.0

Old Fig 7C and New Fig. 8 C+D: differ, but it is unclear what has been changed. In the new version there are obviously less measurements. It seems that the authors have removed data instead of remeasuring them.

We thank the referee for noting this discrepancy. We had originally planned to include data on a homolog of HiSiaP (Comparison #1) called VcSiaP. However, we removed the data before the very first submission since parts of the EPR data were already published, yet unfortunately forgot to delete the datapoints from the plots in the old Figure 7C. The Figure 8CD in the first revision and in the current version (2nd revision) contains only data that is presented and was measured in the manuscript. The data can now be backtracked (see above).

Minor comments:

Page 3 (Line 28-31): The description of PDB-dev is not appropriate. Please have a look at their webpage for a more appropriate description "PDB-Dev is a prototype archiving system for structural models obtained using integrative or hybrid modeling and is funded by the NSF ABI Development Program".

This was changed as follows: "The hybrid models produced by such integrative approaches can be deposited in the PDB-Dev database⁹."

Reviewers' Comments:

Reviewer #2:

Remarks to the Author:

I see that authors improved the manuscript significantly. However, some improvements with respect to fluorescence spectroscopy are still insufficient.

1.) One prime example is the management and display of measurement errors. While the errors of EPR are displayed in each figure as read shaded areas, no error-bars are shown for fluorescence. Instead, the values are "hidden" in the SI. The authors should add errors bars for fluorescence in all figures of the main text so that the displays are balanced.

2.) Moreover, the reported fluorescence quantum yields of TMR are questionable and do not agree to the reviewers experience. The authors should consult for example work of Bob Clegg where TMR was used in nucleic acids where quenching is usually much stronger (because of G). In proteins, the fluorescence quantum yields of TMR are even higher. The authors should either measure the fluorescence brightnesses or fluorescence lifetimes of the conjugates and normalize them (option 1) or they should define a proper range of uncertainty of R_0 (see book of Wieb van der Meer), by stating they did/could not measure the fluorescence quantum yield thus the potential fluorescence quantum yields can range from 0.9 to 0.1 (option 2) and hence conclude a large uncertainty. In terms of the semi-quantitative cross validation of EPR and FRET anticipated by the authors option 1 is clearly preferable.

3.) The reviewer appreciates that the authors compared different fluorophores. However, the results are only displayed in the SI. The journal allows the authors to present more display items so that these nice results can be shown the main text.

4.) Figures 8C+D are a good place to add predicted uncertainty ranges (see Hellenkamp) for both methods and compare/discuss the spread of the data with the predictions.

Reviewer #2 (Remarks to the Author):

I see that authors improved the manuscript significantly. However, some improvements with respect to fluorescence spectroscopy are still insufficient.

We would like to thank the referee again for the time and effort invested to review our paper. We are grateful for the constructive input and value the contribution made to improve the quality of the manuscript, which we now hope is suitable for publication.

1.) One prime example is the management and display of measurement errors. While the errors of EPR are displayed in each figure as read shaded areas, no error-bars are shown for fluorescence. Instead, the values are "hidden" in the SI. The authors should add errors bars for fluorescence in all figures of the main text so that the displays are balanced.

We have repeated all smFRET experiments to provide error bars / standard deviation of the mean distances (from at least 3 independent experiments) and incorporated error bars into the respective figures. The results of all individual measurements are listed in SI Table 6.

SI Table 6: Experimental distances of three independent smFRET measurements.

Sample	FRET Exp 1 (year) [Å]	FRET Exp 2 (year) [Å]	FRET Exp 3 (year) [Å]	FRET Exp mean [Å]	Std. Dev. [Å]
HiSiaP 58/134 apo	62.4 (2019)	64.6 (2021)	65.9 (2019)	64.3	1.8
HiSiaP 58/134 holo	57.8 (2019)	58.2 (2021)	59.4 (2019)	58.5	0.8
HiSiaP 55/175 apo	42.5 (2019)	40.3 (2021)	41.6 (2019)	41.5	1.1
HiSiaP 55/175 holo	38.9 (2019)	36.9 (2021)	38.4 (2019)	38.1	1.0
HiSiaP 112/175 apo	55.1 (2019)	51.3 (2021)	52.1 (2022)	52.8	2.0
HiSiaP 112/175 holo	56.9 (2019)	53.3 (2021)	53.1 (2022)	54.4	2.1
HiSiaP 175/228 apo	59.6 (2019)	55.3 (2021)	55.4 (2022)	56.8	2.4
HiSiaP 175/228 holo	60.6 (2019)	56.9 (2021)	56.3 (2022)	57.9	2.4
MalE 134/186 apo	39.5 (2018)	40.9 (2022)	41.7 (2018)	40.7	1.1
MalE 134/186 holo	40.1 (2018)	40.9 (2022)	41.9 (2018)	41.0	0.9
MalE 29/352 apo	63.7 (2018)	62.7 (2018)	65.1 (2018)	63.8	1.2
MalE 29/352 holo	56.6 (2018)	55.8 (2018)	57.9 (2018)	56.8	1.1
MalE 36/352 apo	55.2 (2018)	54.6 (2022)	53.5 (2022)	54.4	0.9
MalE 36/352 holo	45.6 (2018)	44.8 (2022)	44.1 (2022)	44.8	0.8
MalE 87/127 apo	43.2 (2018)	41.6 (2022)	42.4 (2022)	42.4	0.8
MalE 87/127 holo	48.9 (2018)	47.4 (2022)	48.4 (2022)	48.2	0.8
SBD2 319/392 apo	54.7 (2018)	54.3 (2018)	52.8 (2022)	53.9	1.0
SBD2 319/392 holo	46.1 (2018)	45.0 (2018)	44.4 (2022)	45.2	0.8
SBD2 369/451 apo	60.5 (2018)	60.3 (2018)	59.5 (2022)	60.1	0.5
SBD2 369/451 holo	52.5 (2018)	51.8 (2018)	51.0 (2022)	51.8	0.7
YopO 113/497 apo	58.3 (2021)	57.1 (2021)	56.7 (2022)	57.3	0.8
YopO 113/497 holo	69.0 (2021)	68.7 (2021)	67.2 (2022)	68.3	1.0

2.) Moreover, the reported fluorescence quantum yields of TMR are questionable and do not agree to the reviewers experience. The authors should consult for example work of Bob Clegg where TMR was used in nucleic acids where quenching is usually much stronger (because of G). In proteins, the fluorescence quantum yields of TMR are even higher. The authors should either measure the fluorescence brightnesses or fluorescence lifetimes of the conjugates and normalize them (option 1) or they should define a proper range of uncertainty of R0 (see book of Wieb van der Meer), by stating they did/could not measure

the fluorescence quantum yield thus the potential fluorescence quantum yields can range from 0.9 to 0.1 (option 2) and hence conclude a large uncertainty. In terms of the semi-quantitative cross validation of EPR and FRET anticipated by the authors option 1 is clearly preferable.

We thank the reviewer for insisting on this point. We selected option 1 and determined the quantum yields for TMR on HiSiaP in comparison to Rhodamine 6G and free TMR. The experiments are presented in SI Figure 22 and all additional parameters for determination of the Förster radius are summarized in SI Table 5

SI Figure 22: Quantum yield measurement of TMR in reference to Rhodamine 6G. A) Absorbance spectra of Rhodamine 6G (top), TMR (middle), HiSiaP-175-TMR in apo and holo

states (bottom) at 6 different concentrations. The dashed line indicates the extracted absorbance values from the excitation wavelength at 510 nm. **B)** Emission spectra of Rhodamine 6G (top), TMR (middle), HiSiaP-175-TMR in apo and holo states (bottom) of the sample in (A) excited at 510 nm. **C)** The integrated emission from (B) is plotted against the absorbance extracted from A and fitted to the function $I_{\text{int}} = m \cdot A$. Triplicates were used to determine the quantum yield (QY).

SI Table 5: Förster radius calculation for Alexa Fluor 555 – Alexa Fluor 647 and TMR-Cy5.
Overview of all used parameters (measured/literature).

Pair	Alexa Fluor 555 – Alexa Fluor 647	TMR – Cy5
Orientation factor κ^2	2/3	2/3
Average refractive index	1.4	1.4
Extinction coefficient	265,000 1/(M cm)	250,000 1/(M cm)
Quantum yield donor	0.14	0.395*
Overlap integral	$8.12 \times 10^{15} \text{ nm}^4 / (\text{M cm})$	$1.13 \times 10^{16} \text{ nm}^4 / (\text{M cm})$
R₀	51 Å	62.7 Å

*Determined experimentally, SI Figure 22

3.) The reviewer appreciates that the authors compared different fluorophores. However, the results are only displayed in the SI. The journal allows the authors to present more display items so that these nice results can be shown the main text.

While more display items are allowed, we decided to leave the figure in the SI. The data are very interesting, but we think it distracts from the main story line, which is a comparison between PELDOR/DEER and smFRET.

4.) Figures 8C+D are a good place to add predicted uncertainty ranges (see Hellenkamp) for both methods and compare/discuss the spread of the data with the predictions.

This is a good idea and we added the uncertainty ranges to the two panels. It is also mentioned in the first paragraph of the discussion text: “The spread of the datapoints agrees well with the error margins of $\pm 3.5 \text{ Å}$ and $\pm 5 \text{ Å}$ that are commonly given for PELDOR/DEER and smFRET, respectively (grey shades in Figure 8CD)^{97 65–67,91}. Within the dataset, no systematic offset was observed between PELDOR/DEER or smFRET. However, Figure 8C indicates a trend, where longer distances show a larger deviation in the prediction accuracy for both methods.”

Figure 8: Comparison of PELDOR/DEER and smFRET measurements and the influence of linker length on the correlation between experimental and predicted distances. **A**) Multiple ensembles of spin- and fluorescence labels were simulated with mtsslWizard using the open form of HiSiaP (PDB-ID: 2CEY²⁹) and the labelling sites 55, 58, 112, 134, 175 and 225. In the schematic, the radius of the sphere represents the length of the linker that connects the fluorophore or spin center to the C-alpha atom of the labelled residue. Interactions with the protein surface (grey arcs) are indicated and lead to a clustering of labels at that position. Depending on the degree of interaction between protein and label, the accessible volume approach becomes less accurate. **B**) Histograms of 1000 simulations described in A) with a 10 Å linker (upper plot) and 20 Å linker (lower plot) and varying degree of protein label interaction. The percentage indicates how many percent of the 1000 dummy atoms are localized at the interaction site. As example for a long (20 Å) and immobilized linker, the protein structure of MMP-12 (matrix metalloproteinases, PDB-ID 5L79⁹⁶) in conjugation with a Cy5.5 fluorophore (K241, colored spheres) was selected. The surface of the protein is shown in grey and the accessible volume of the fluorophore, calculated with FPS⁶⁷ is shown as a blue mesh. The dark- and light-grey shades represent the error margins of ± 3.5 Å and ± 5.0 Å that are often given for PELDOR/DEER and smFRET experiments, respectively. **C**) Predicted vs experimental smFRET or PELDOR/DEER distances of datasets that were measured with both methods in this study. **D**) As C) but the differences are plotted against the experiment number in SI Table 2. **E**) Comparison of the distances determined by PELDOR/DEER or smFRET and the simulation for the same experiment.